# Improved human greenspace exposure equality during 21st century urbanization

Shengbiao Wu ◎[1], Bin Chen ◎[1,2,3] ✉, Chris Webster ◎[2,3,4], Bing Xu[5] & Peng Gong ◎[2,6,7]

Greenspace plays a crucial role in urban ecosystems and has been recognized as a key factor in promoting sustainable and healthy city development. Recent studies have revealed a growing concern about urban greenspace exposure inequality; however, the extent to which urbanization affects human exposure to greenspace and associated inequalities over time remains unclear. Here, we incorporate a Landsat-based 30-meter time-series greenspace mapping and a population-weighted exposure framework to quantify the changes in human exposure to greenspace and associated equality (rather than equity) for 1028 global cities from 2000 to 2018. Results show a substantial increase in physical greenspace coverage and an improvement in human exposure to urban greenspace, leading to a reduction in greenspace exposure inequality over the past two decades. Nevertheless, we observe a contrast in the rate of reduction in greenspace exposure inequality between cities in the Global South and North, with a faster rate of reduction in the Global South, nearly four times that of the Global North. These findings provide valuable insights into the impact of urbanization on urban nature and environmental inequality change and can help inform future city greening efforts.

Urban areas are the hub of human society, characterized by agglomeration, industrialization, and modernization[1]. Currently, over half of the world's population resides in urban areas and this number is expected to increase to 68% by 2050[2]. While urbanization has brought about many social and economic benefits, such as improved access to public infrastructure, sanitation, and education services[3], it has also resulted in severe environmental degradation, including deforestation[4], habitat loss[5], air and noise pollution[6], and increased fossil fuel consumption and $CO_2$ emissions[7]. This has led to environmental injustice, with some communities bearing a disproportionate share of the negative impact of urbanization[8,9]. Achieving equality in urban environmental sustainability is crucial, as it will help address the imbalance between social, environmental, and economic

developments and reduce society's vulnerability to risks[10]. This is the focus of the 11th Sustainable Development Goal of creating inclusive, resilient, and sustainable cities and human settlements[11].

Greenspace, as a key component of urban nature, offers a wide range of ecosystem services and health benefits, making it a useful proxy for evaluating urban environmental sustainability[12]. Providing universal access to green and public space has been acknowledged as an important pathway in the pursuit of sustainable and healthy development goals. Using greenspace supply metrics such as greenspace total or per capita supply, previous studies have measured the greenspace supply at both the city and global scales[13,14]. Human exposure to greenspace, measured as the averaged amount of greenspace coverage within people's nearby environment expressed as a

[1]Future Urbanity & Sustainable Environment (FUSE) Lab, Division of Landscape Architecture, Department of Architecture, Faculty of Architecture, The University of Hong Kong, Hong Kong SAR, China. [2]Urban Systems Institute, The University of Hong Kong, Hong Kong SAR, China. [3]HKU Musketeers Foundation Institute of Data Science, The University of Hong Kong, Hong Kong SAR, China. [4]HKUrbanLabs, Faculty of Architecture, The University of Hong Kong, Hong Kong SAR, China. [5]Department of Earth System Science, Ministry of Education Ecological Field Station for East Asian Migratory Birds, and Institute for Global Change Studies, Tsinghua University, Beijing 100084, China. [6]Department of Geography, and Department of Earth Sciences, The University of Hong Kong, Hong Kong SAR, China. [7]Institute for Climate and Carbon Neutrality, The University of Hong Kong, Hong Kong SAR, China. ✉e-mail: binley.chen@hku.hk

percentage[15,16], has also been quantified in sampled cities[17–19]. Recent research has used high-resolution satellite imagery and population data, combined with environmental exposure models, to reveal the disparities in greenspace exposure between cities in the Global North and Global South. Urban residents in Global South cities (i.e., cities of developing countries, such as the Chinese, Indian, and Middle Eastern cities) have only one-third the exposure to greenspace compared to those in Global North cities (i.e., cities of developed countries, such as United States, European, and Australian cities), but experience twice the level of inequality[17]. However, existing studies on greenspace exposure and inequality are constrained to data from individual baseline years that cannot capture the impacts of long-term urban development[20,21] and the gradual temporal evolution of human-nature interactions such as urban warming and greenspace management strategies[22,23], limiting our understanding of the impact of urban development on greenspace supply, human exposure, and inequality over time.

The impacts of urbanization on greenspace exposure are of two kinds:[24,25] the replacement of green land cover with built land cover, and the enrichment of built land cover with designed green land. This replacement often occurs during early-stage urban development via rapid urban expansion, transforming naturally vegetated areas into artificial impervious surfaces and ultimately resulting in a subsequent decrease in greenspace coverage. Historically, there is typically a process of greenspace destruction as agricultural and natural green land is replaced by impervious surfaces and then a later process of re-greening via environmental improvements. Urban expansion through comprehensive spatial planning (as opposed to incremental unplanned sprawl), tends to 'reprovision' greenspace by design from the outset, but these spaces themselves tend to evolve (decrease and increase in quantity and quality) over time according to income, land value and environmental preferences[26,27]. In later periods, a highly urbanized city generates its own ecology, with factors such as urban warming and $CO_2$ fertilization leading to an extension of the growing season and increased greenspace growth[24,28,29]. City management practices, such as planting street trees and creating vertical gardens, can also increase the overall greenspace supply in an urban environment developed over time under a legal environmental planning regime[30,31]. The growth of the urban population modifies the interactions between the population and the urban green environment, leading to changes in human exposure to greenspace[16–18,32]. Greenspace is a 'superior good', with demand increasing as the prosperity of citizens increases[33]. In contrast to a crowding effect of population growth, leading to reduced greenspace exposure, an income effect tends to increase greenspace supply, diversity, and aesthetics in wealthier cities. The two effects (destruction and construction of greenspace) can also occur periodically and are not always strictly chronological. Different parts of cities developed under different political and economic conditions often display different patterns of greenspace, for example, highly planned systems of public local parks in the post-World War II cities; intensive street-tree planting during the first half of the 20th century; private greenspaces in ancient city quarters, very little greenspace in a period of informal urban expansion[34–37].

While both destructive and constructive processes are well understood in the context of particular cities and professional urban management activities[38,39], limited studies have investigated the net change in greenspace supply and human exposure to greenspace across a global sample of cities for a comparable time, to identify general trends. Such trends are the net outcomes of the destructive and constructive processes often at work simultaneously in the evolution of city fabric. Inequality in greenspace exposure is an increasing concern as it can be translated into adverse effects in mental and physical health[18,19]. Individual studies show how greenspace provision and the joint effects of greenspace provision and spatial configuration control greenspace exposure inequality[17]. However, the drivers of changing greenspace exposure inequality over time remain unclear and limit our understanding of the relationship between greenspace and population distributions in determining and attempting to promote greenspace exposure equality in the future projection.

Here we generate a global urban greenspace dataset for 1028 large cities (i.e., urban area > 100 km$^2$) using 30-m-resolution Landsat satellite data from 2000 to 2018, and validate its accuracy using the 1-m-resolution National Agriculture Imagery Program (NAIP) aerial data and 10-m Sentinel-2 satellite observations. Specifically, we adopted a broad definition of greenspace to refer to land that is partly or completely covered with grass, trees, shrubs, and other vegetation[32]. That is, green-covered land within the curtilage of an urban boundary, since we are focusing on cities in this study. We further leveraged the 30-m-resolution satellite-derived greenspace coverage map (i.e., the vertical projection area of greenspace within a $30 \times 30$ m$^2$ pixel) and 100-m population data, together with the population-weighted exposure model, to assess the spatiotemporal trends of urban greenspace exposure, inequality, and the associated drivers over the first two decades of 21$^{st}$ century. The equality term in this study is to quantify whether individual people share the even greenspace resource[40]. To this end, we included cities with different urban development stages, classified as Global North and Global South cities, by answering the following four questions. (1) What has been the fate of urban greenspace supply under global urbanization during the first two decades of the current century? (2) Has human exposure to greenspace and the inequality issue improved or worsened in this period? (3) What are the relative contributions of greenspace provision and population growth to changes in exposure and the associated inequality? (4) What can be learned from the start of the century to guide city greening in the decades ahead?

## Results
### Spatiotemporal pattern of physical greenspace coverage
We used the 30-m-resolution Landsat-derived fractional greenspace mapping to quantify the temporal change of physical greenspace coverage for 1028 global cities from 2000 to 2018 (see Methods). Results reveal a contrasting pattern in the direction of temporal change in greenspace coverage between Global North and South (Fig. 1a). Global North cities show a prominently increasing trend in greenspace coverage, including Europe, North America, Russia, Australia, and a few cities of East Asia. Global South cities have a decreasing trend, most of which are spatially clustered in East Asia, Southeast Asia, Africa, and Latin America. Global North cities experienced a higher trend in the absolute magnitude of greenspace exposure level ($2.58 \times 0.001$ yr$^{-1}$) than Global South cities ($-2.51 \times 0.001$ yr$^{-1}$), resulting in an overall slightly positive change trend for all global cities ($0.22 \times 0.001$ yr$^{-1}$) (Table 1). By continent, European cities have the largest level of change trend for greenspace exposure ($4.41 \times 0.001$ yr$^{-1}$), which is twice that of Asian ($-2.11 \times 0.001$ yr$^{-1}$) and Australian and Oceanian ($2.16 \times 0.001$ yr$^{-1}$) cities. Cities in South America ($-0.44 \times 0.001$ yr$^{-1}$) and Africa ($0.51 \times 0.001$ yr$^{-1}$) experience the lowest temporal trend level, which is approximately half of that of the North American cities ($0.95 \times 0.001$ yr$^{-1}$). Interestingly, we find a turning point around the year 2011 in the temporal trajectory of annual greenspace coverage for Global North and Global South cities (Fig. 1b). Before this turning point, Global North cities have a slight reduction in annual greenspace coverage, while Global South cities are sharply losing greenspace coverage in this period. After 2011, both Global North and Global South cities experienced an increasing annual greenspace coverage, with a larger magnitude observed for Global North cities. Continents show different directions and magnitudes in the temporal trajectories of annual greenspace coverage (Supplementary Fig. 1a). Our sensitivity analysis regarding the greenspace mapping with the spectral unmixing-based threshold classifications (see Methods) show very similar temporal evolution patterns of

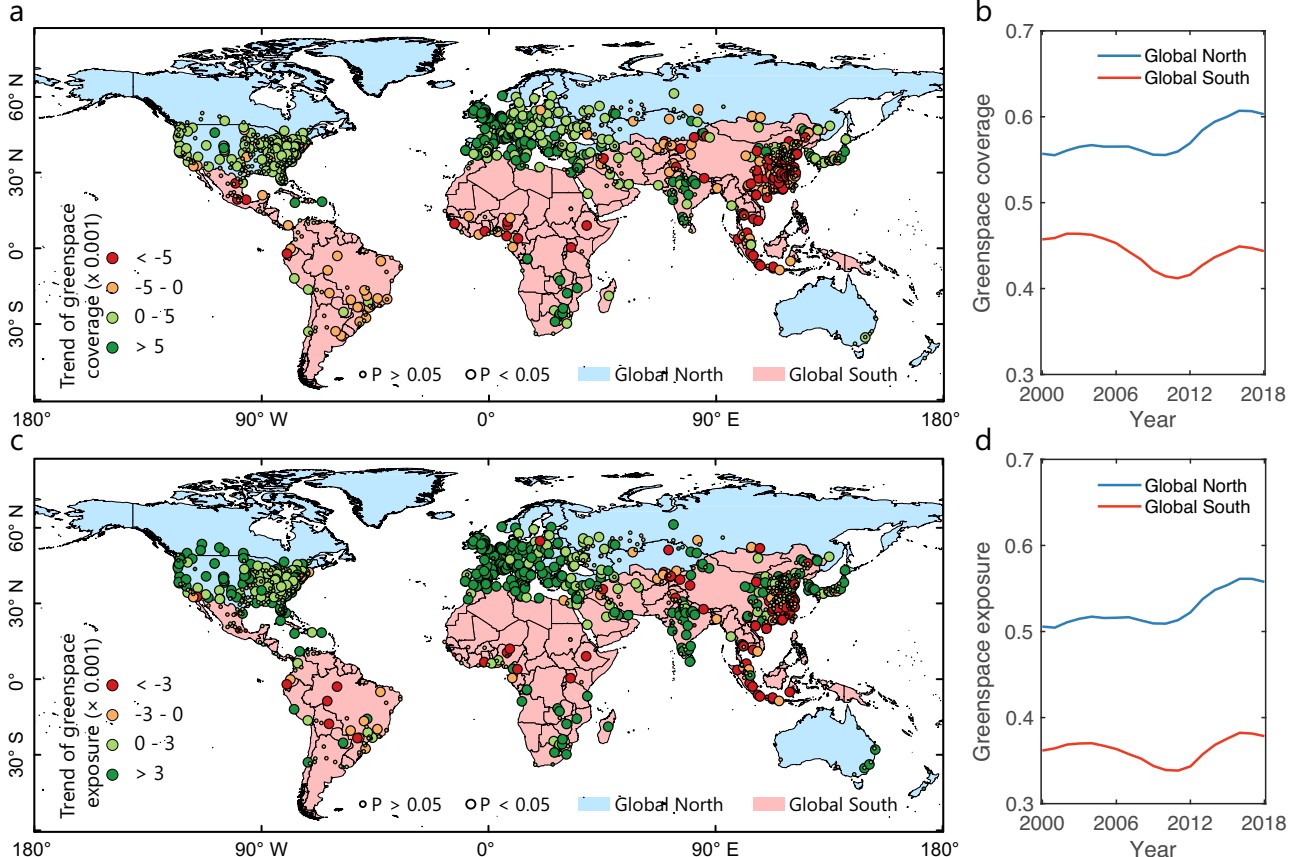

**Fig. 1 | Temporal changes of physical greenspace coverage (GC) and green-space exposure (GE) for global 1028 cities from 2000 to 2018. a** City-level temporal trend of GC changes. **b** Annual GC dynamics for Global North and Global South cities. **c** City-level temporal trend of GE changes. **d** Annual GE dynamics for Global North and Global South cities. In **a** and **c**, the city-level temporal change trends are divided into four qualitative levels, where cool (dark and slight green) and warm colors (yellow and red) refer to increasing and decreasing greenspace trends, respectively. The non-parametric Theil–Sen slope estimator approach is used to determine the long-term trends of both GC and GE. The non-parametric Mann–Kendall is used to evaluate the significance of these detected temporal trends. Large bubble sizes represent a statistically significant level of 0.05 (p-value < 0.05) and small bubble sizes represent a non-significant trend with p-value > 0.05. The administrative boundaries data is from the Global Administrative Areas (GADM) (https://gadm.org/).

**Table 1 | Statistics of temporal trends of city-level greenspace coverage, human exposure to greenspace, and greenspace exposure inequality across regions**

| Region (# of cities) | Greenspace coverage (×0.001 yr⁻¹) | Greenspace exposure (×0.001 yr⁻¹) | Gini of greenspace exposure (×0.001 yr⁻¹) |
|---|---|---|---|
| **Global North (522)** | **2.58 ± 3.04** | **3.10 ± 2.37** | **−0.94 ± 1.05** |
| **Global South (506)** | **−2.51 ± 4.68** | **0.44 ± 3.70** | **−3.54 ± 3.53** |
| North America (293) | 0.95 ± 2.20 | 2.25 ± 1.84 | −0.99 ± 1.65 |
| South America (60) | −0.44 ± 2.51 | −0.21 ± 2.07 | −0.85 ± 1.77 |
| Europe (180) | 4.41 ± 3.78 | 4.64 ± 2.58 | −1.12 ± 0.97 |
| Africa (63) | 0.51 ± 3.41 | 1.64 ± 3.61 | −2.09 ± 3.48 |
| Asia (420) | −2.11 ± 5.07 | 0.55 ± 3.77 | −3.80 ± 3.41 |
| Australia/Oceania (12) | 2.16 ± 1.66 | 2.55 ± 1.74 | −0.82 ± 0.43 |
| Global (1028) | 0.22 ± 4.64 | 1.79 ± 3.37 | −2.22 ± 2.89 |

greenspace coverage (Supplementary Figs. 2 and 3, and Supplementary Tables 1–3).

## Spatiotemporal patterns of greenspace exposure and associated inequality

To adjust gross greenspace supply to reflect assumed demand, we used population distribution maps and a population-weighted exposure framework to quantify the temporal change of human greenspace exposure and inequality. Like greenspace coverage, human exposure

to greenspace shows contrasting temporal change trends between Global North and Global South cities (Fig. 1c). While the global cities have an increasing greenspace exposure (1.79 × 0.001 yr⁻¹) (Table 1), Global North cities experience a much higher temporal change magnitude (3.10 × 0.001 yr⁻¹) than Global South cities (0.44 × 0.001 yr⁻¹). In North American, European, African, Australian, and Oceanian cities, population distribution amplifies the share of greenspace coverage, with a greater temporal change magnitude in human greenspace exposure compared to physical greenspace coverage. An important

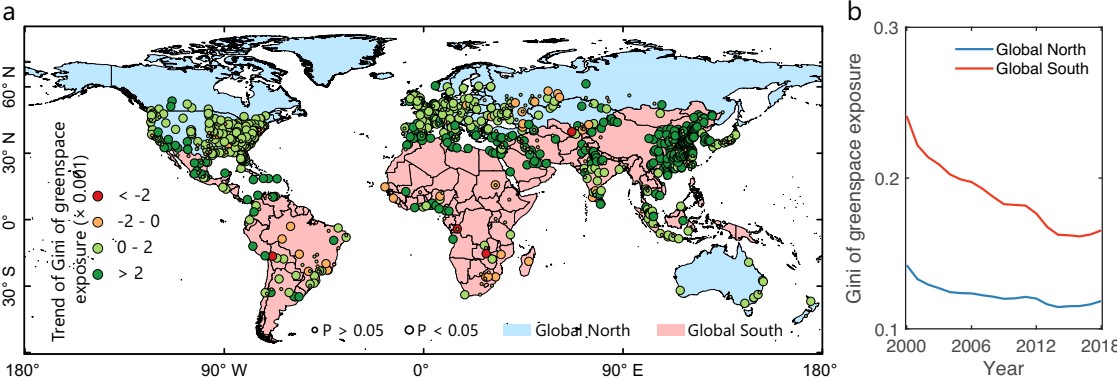

**Fig. 2 | Temporal change of greenspace exposure inequality measured by the Gini index for global 1028 cities from 2000 to 2018. a** City-level temporal trend of Gini index. **b** Annual dynamics of the Gini index on average for Global North and Global South cities. Similarly, large bubble sizes represent a statistically significant level of 0.05 ($p$-value < 0.05) and small bubble sizes represent a non-significant trend with $p$-value > 0.05. The administrative boundaries data is from the Global Administrative Areas (GADM) (https://gadm.org/).

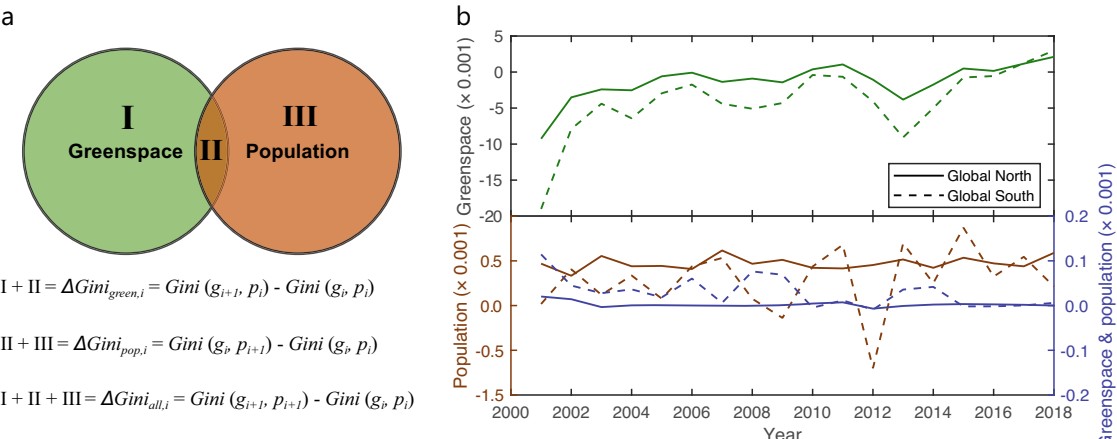

**Fig. 3 | Attribution of drivers accounting for temporal change in greenspace exposure inequality. a** The Venn conceptual model of quantifying individual and joint contributions from greenspace and population to the overall change in the Gini index. Green circle (regions I and II) denotes the Gini index change $\Delta Gini_{green,i}$ induced by the change of greenspace provision from $g_i$ to $g_{i+1}$ with a fixed population $p_i$ for year $i$ ($i$ = 2001,..., 2018). Orange circle (regions II and III) denotes the Gini index change $\Delta Gini_{pop,i}$ induced by the population change from $p_i$ to $p_{i+1}$ with a fixed greenspace $g_I$ for year $i$. These two circles (regions I, II, and III) denote the overall change of Gini index $\Delta Gini_{all,i}$ induced by both greenspace and population change from ($g_i$, $p_i$) to ($g_{i+1}$, $p_{i+1}$) for year $i$. The overlapped region II denotes the joint effects of greenspace and population changes. **b** Individual and joint effects of greenspace and population changes to the temporal dynamics of the Gini index from 2000 to 2018, grouped by Global North and Global South cities.

finding is that despite a negative trend of greenspace coverage in Global South and Asian cities, people in these cities experienced a slightly positive change in greenspace exposure in the first two decades of the 21st century. Annual continental-level greenspace exposure has similar temporal trajectory patterns as greenspace coverage but shows larger variabilities across continents (Supplementary Fig. 4). Sensitivity analysis of the spectral unmixing-based threshold classifications shows that these findings are consistent for different greenspace mapping methods (Supplementary Figs. 5 and 6, and Supplementary Tables 1–3). Our results show a dominantly decreasing trend in greenspace exposure inequality (measured by Gini index) for global cities, regardless of Global North or South, together with a decreasing annual continental-level average over time (Fig. 2). Global South cities experienced a higher change level of greenspace exposure inequality (−3.54 × 0.001 yr⁻¹), almost four times that of Global North cities (−0.94 × 0.001 yr⁻¹) (Table 1). The changing trends of the Gini index of greenspace exposure for North American, South American, European, African, Asian, Australian, and Oceanian cities are in the range of (−0.85)-(−3.80) × 0.001 yr⁻¹, with a mean value of −2.22 ×

0.001 yr⁻¹ for global cities (Supplementary Fig. 7). These conclusions are consistent with our sensitivity analysis of different greenspace mapping approaches (Supplementary Figs. 8 and 9, and Supplementary Tables 1–3). To further verify these Gini-based results, we adopted the other two widely used economic inequality metrics (i.e., Atkinson and Theil) to measure the inequality of greenspace exposure, which show very consistent results (Supplementary Figs. 10 and 11 and Supplementary Tables 4 and 5).

### Drivers of changing inequality in greenspace exposure

We proposed a Venn conceptual model to examine the drivers of temporal changing trends in greenspace exposure inequality. Taking Gini as an example, results reveal that greenspace coverage, as a measure of greenspace supply, has dominantly promoted the improvement (i.e., negative magnitude) in the temporal change of greenspace exposure inequality over first two decades of the 21st century. The absolute contribution of greenspace coverage to the reduction in the Gini index is larger in Global South cities than Global North cities (Fig. 3 and Supplementary Fig. 12). We also find that these

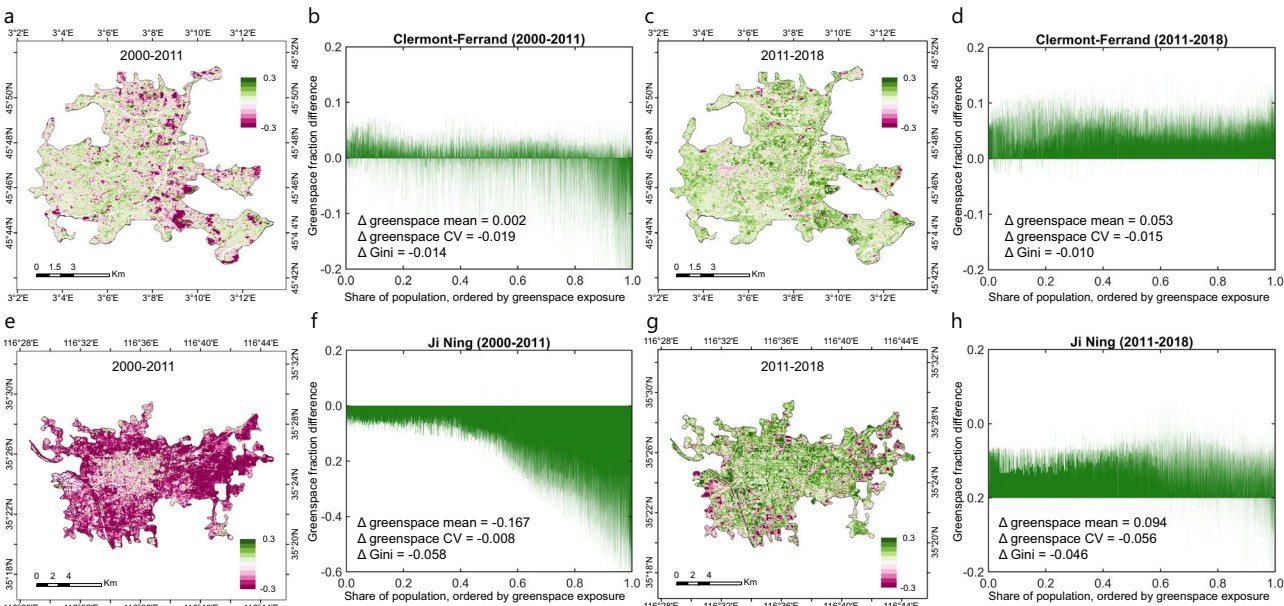

**Fig. 4 | City-level illustrative examples of greenspace changes modifying the variations of greenspace exposure inequality for Clermont-Ferrand, France in the Global North (a-d) and Jining, China in the Global South (e-h). a** and **e** The spatial pattern of greenspace coverage (GC) changes from 2000 to 2011. **b** and **f** GC change over individual population from 2000 to 2011. **c** and **g** Spatial pattern of GC change from 2011 to 2018. **d** and **h**. GC change over individual population from 2011 to 2018. CV refers to the coefficient of variation for individual exposure to greenspace.

greenspace coverage-induced improvement effects are loosening over time during this period. Compared to greenspace coverage, the population distribution and its combined effects with greenspace coverage, have slighter contributions to the temporal change of Gini. In addition to results based on the Gini index, our sensitivity analysis using the Atkinson and Theil metrics also show very similar attribution results (Supplementary Figs. 13 and 14). We also incorporated the comparative attribution approach[41] to verify the robustness of the proposed Venn conceptual model, similar results show that the contribution from greenspace dominates the overall change in the Gini index compared to population growth (Supplementary Fig. S15). To dive into the controls of greenspace coverage on the temporal change of the Gini index, we selected two cities with contrasting greenspace coverage change patterns as pilots, Clermont-Ferrand city, France, in the Global North and Jining city, China, in the Global South. Clermont-Ferrand city has a net-balanced greenspace coverage change from 2000 to 2011 but a substantial increase from 2011 to 2018 (Fig. 4a, c). Greenspace coverage in Jining City shows a notable reduction from 2000 to 2011 and recovers around half of the previous level from 2011 to 2018 (Fig. 4e, g). Results reveal that the regulations of greenspace coverage on greenspace exposure inequality have different pathways. When greenspace provision increases, the opportunities of being exposed to urban greenspace resources will be improved for individual urban residents, and thereby reducing the greenspace exposure inequality (Fig. 4d, h). By contrast, if greenspace provision decreases, or remains stable in the face of population growth, urban residents initially being highly exposed to greenspace might lose greenspace exposure, resulting into a more balanced greenspace share (at a lower exposure level) (Fig. 4b and f). These regulating pathways are supported by a reduction in the coefficient of variation (CV) which can directly measure the relative dispersion of greenspace share among resident individuals within the city.

## Discussion

Global urbanization promotes increasing population growth and socio-economic prosperity but also enhances environmental degradation, which substantially challenges urban sustainability[42,43]. The

unequal distribution of greenspace and its exposure is one of the pressing issues in urban sustainability and public health, especially in rapidly urbanizing Global South cities[44]. Greenspace, as a key component of urban nature and a direct pathway to achieving the 11th Sustainable Development Goal, is widely understood to be reduced by urbanization for individual and even global cities[17,32]. We note, however, this is not entirely true, due to greenspace being a superior economic good (i.e., good beyond the routine part of everyday life), the demand for which rises with income[45,46]. This gives rise to different dynamics between the Global South and North, and differences within cities of these two broad global regions.

Recent satellite-based evidence reveals that at the start of the 21st century, the positive effects of urbanization on vegetation growth have been increasing, which may partially counteract the loss of vegetation caused by land transformation in earlier stages of urbanization, and lead to improved human exposure to greenspace[24,25]. By combining this data with global population mappings, we were able to quantify the temporal trends in greenspace exposure and inequality over the past two decades. Our results show that greenspace coverage has increased overall across global cities (Fig. 1a), with 648 (63%) out of 1028 global cities experiencing positive trends of vegetation increases. This suggests that greening management activities have surpassed the loss of vegetation caused by land cover transformation in urban expansion. This finding is consistent with results from previous studies based on coarse-resolution satellite observations over global areas[47–49] and city-specific regions[25,50]. The rate of urban greening varies based on factors such as climate, urban development intensity, and population density[25].

Prominent spatial differences in urban greenspace trends have been revealed across our sample of global cities, particularly in the contrast between cities in the Global North and Global South. Global North cities that are highly urbanized have become greener over the past two decades, the likely cause of which is more green real estate building and green urban planning and city management by both private and public developers and agencies[51]. On the other hand, cities with lower levels of urbanization in the Global North are still experiencing vegetation cover loss (Fig. 1a). This contrast highlights the different priorities for greening management policy and action in

Global North and Global South cities. Global North cities might encourage optimizing greenspace provision configuration and quality to maximize greening benefits (e.g., climate cooling)[52,53] and minimize risks (e.g., pollen allergy)[54,55]. Global South cities may need to increase green planting programs to make up for the reduction in physical greenspace coverage[17]. Interestingly, a turning point in the trend of greenspace provision and exposure was observed around the year 2011, when the world's urban population exceeded the rural population[56] and natural and anthropogenic activities plausibly related to a net global vegetation greening (e.g., $CO_2$ fertilization, climate change, land-use change, and nitrogen deposition) have been enhanced[47]. Prior to this, cities globally, were experiencing a decline in greenspace coverage, but after 2011, the trend reversed turning to a positive increase, indicating a net greenspace provision.

Global cities, represented by our 1028 samples experienced a net increase in exposure to greenspace during the start of this century (Fig. 1b). With the exception of cities in South America, all regions in the world, namely, cities in North American, European, African, Asian, Australian, and Oceanian, experienced increased greenspace (Table 1) over this period. Compared with greenspace coverage, greenspace exposure has changed with a larger magnitude due to greenspace-human interactions. The trend occurred through urbanization-induced vegetation loss in the outer belts of cities with a much lower population[57] and concurrently, greenspace coverage increases and population grows in new urban areas[16]. Our findings show a substantial difference in greenspace exposure change rates between cities in the Global North and Global South, with cities in the Global North experiencing approximately seven times for the increase rate of greenspace exposure compared to those in the Global South (Table 1).

Our analysis also reveals a trend toward equality in human exposure to greenspace, as reflected by a decline in inequality indices such as the Gini, Atkinson, and Theil coefficients (Fig. 2). Based on the Venn conceptual attribution model, we identify that the primary driver behind this reduction in inequality is the provision of greenspace (Fig. 3), which operates through two different pathways. In early stages, greenspace loss caused by urban expansion tends to disproportionately affect populations with high levels of greenspace exposure, reducing disparities in exposure among the population. However, as urbanization continues, the increase of greenspace is typically much more equally spread over the population, which shares a city's particular level of greenspace exposure and thus narrows the gap in greenspace exposure across population (Fig. 4). Our results provide evidence and offer valuable insights to help government agencies, city planners, and private developers adopt holistic urban development strategies to improve the amount and quality of greenspace supply to achieve sustainable development goals. The study tells a broadly positive story of the opening decades of the 'urban century', and our analysis of trends, and subsequent studies of positive outliers in those trends, will help cities achieve better net outcomes when planning for balanced changes in urban greenspace loss and construction by incorporating multidimensional contexts of greening history, greenspace supply status quo, prioritized vulnerable hotspots and the underlying socio-economic factors. The power of big-picture global narrative studies to influence policy-makers, national debates, and lobbyists, should also not be underestimated. Consistent with our net-effects analysis, we encourage cities to take a greenspace exposure balance-sheet approach to target SDGs for their populations, using a high-resolution sensing technique as we used.

There are also some levels of limitations that should be acknowledged in this study. First, although the integration of 30-m resolution Landsat satellite with a spectral unmixing approach can help resolve the sub-pixel greenspace mapping, our method is still unable to explicitly identify certain greenspace types in small and fragmented patches at the Landsat's resolution, such as street plantation, lawns, and pocket gardens and parks, given the heterogeneous

landscape of cities. Therefore, satellites with a higher spatial resolution such as 10-m Sentinel-2, 3-m PlanetScope, and sub-meter WorldView imageries can be incorporated to offer advanced observational opportunities for urban greenspace mapping, but one of the key challenges is how to develop robust spatiotemporal reconstruction algorithms for generating high-quality historical archives. Second, the spatial distributions of greenspace and population footprints within each year of this study are static without modeling spatiotemporal interactions between greenspace and humans dynamically. On the one hand, people living in cities are spatially moving in their daily routines, being exposed to different nearby green environments beyond those near to their place of residence. On the other hand, seasonal changes in urban greenspace that reflect different phenological phases will influence the greenspace availability over time. Therefore, a promising open topic for follow-up research is to integrate a human mobility dataset with dynamic greenspace observations to enable a spatially and temporally explicit human-greenspace interaction framework and move toward real-time greenspace exposure assessment. Third, a global sample of 1028 cities is adopted to investigate urban greening and greenspace exposure inequality across a large geographical spectrum, allowing us to make conclusions that transcend the local observations that limit many greenspace studies. We acknowledge, however, that the area criterion of 100 km² excludes small and medium-sized cities, which are home to millions of people. It will be of interest to compare our results for spatially large cities with a replicated study of smaller cities, since cities of different sizes use space distinctly and have different socio-economic functions and greenspace scales with city size in predictable ways[58]. These unique processes, in turn, influence the greenspace supply-demand relationship and associated temporal development trajectory, further underscoring the importance of this analysis. One future research direction is to explore the associations of a wider spectrum of urban characteristics (e.g., size, shape, density, function, and socio-economic structure) with greenspace, and measure specific human-centric exposures. Clearly, another important factor in assessing the degree and distribution of access or exposure to greenspace is demographic structure such as gender, age, income, and race. In the present study, greenspace exposure differences across population groups are not considered, since their measurement and a technical consideration of between-group equity is beyond the scope of our methods in this study. We therefore talk in terms of equality, rather than equity of exposure and our measurements relate to human population counts abstracted from specific population sub-groups. A deeper understanding of greenspace exposure and demographic structure is becoming socially and politically urgent[17,59], which will help identify 'hotspot' areas with less greenspace that disproportionately affect certain vulnerable populations, and potentially help quantify the variations in health outcomes.

## Methods
### Research design
We generated two-decade time-series greenspace maps from the 30-m-resolution satellite data for global large cities using the linear spectral unmixing approach. We further combined the long-term fine-scale greenspace and population mappings with a population-weighted exposure framework to explore temporal changes in greenspace coverage, human greenspace exposure, and the associated inequality. To decipher the drivers controlling the inequality in human exposure to greenspace over time, we proposed a Venn conceptual model to quantify the individual and joint contributions from greenspace and population for the past two decades. The full workflow of the research design is shown in Supplementary Fig. 16.

### Global urban areas
We selected 1028 major urban areas globally as our study targets. The boundaries of these urban areas were extracted from the latest global

urban boundaries (GUB) product[60]. As the high-quality year-by-year GUB datasets are lacking, we chose the GUB data in 2018 with a geographic area larger than 100 km$^2$ as the urban boundary to allow for 1) the exploration of the urban expansion impacts on greenness change and 2) the collection of sufficient samples for measuring human greenspace exposure and inequality[17]. It is noted that we refer to these 1028 urban areas as 'cities' throughout the manuscript.

## Landsat satellite imagery

We used 19-year (2000–2018) Landsat surface reflectance products from three satellite sensors (i.e., Landsat-5, Landsat-7, and Landsat-8) with a 30-m spatial solution to quantify the spatial distribution and temporal dynamics of greenspace. Landsat provides the longest high-quality temporal record of global surface reflectance data together with the pixel-level quality assurance (QA) layer indicating cloud, cloud shadow, snow, and ice conditions[61]. We used the surface reflectance products of three visible (i.e., blue, green, and red) and one near-infrared band. To minimize the uncertainty caused by Landsat-7 scan line off failure[62], we primarily focused on the use of Landsat-5 and Landsat-8 Collection 2/Tier 1 Level-2 products, with data availabilities of 12-year Landsat-5 (2000–2011), 2-year Landsat-7 (2012-2013), and 5-year Landsat-8 (2014–2018).

Several data pre-processing steps are conducted to ensure high-quality inputs for greenspace mapping. We first harmonized Landsat-5 and Landsat-7 data using Landsat-8 data as a baseline to generate consistent time-series data that removes potential impacts due to the difference in spectral settings across satellite sensors[63]. With the QA bitmask layer, we excluded the pixels that were contaminated by cloud cover, shadow, and snow. We further calculated the normalized difference vegetation index (NDVI) and normalized difference water index (NDWI) to quantify the spectral characteristics of green vegetation and water bodies, respectively. Finally, we adopted a maximum value composite approach to generate the annual greenest vegetation green metrics by selecting the pixel-based maximum NDVI values from the cloud-free time series within a one-year cycle. In addition to the maximum NDVI value, we also recorded the corresponding NDWI and spectral reflectance of blue, green, red, and near-infrared bands for the following fractional greenspace coverage mapping.

## Population data

We used the WorldPop dataset from 2000 to 2018 to map the population's spatially explicit distribution and temporal dynamics. World-Pop provides a global annual update of demographic datasets (e.g., population density, age and sex structures, and urban growth) at a 100-m spatial resolution using a random forest regression tree-based mapping approach[64]. We chose the WorldPop population density dataset for its advantages of high spatial resolution, annual update frequency, and global coverage over alternatives such as Gridded Population of the World (GPW)[65], LandScan[66], and High-Resolution Settlement Layer (HRSL)[67].

## Greenspace coverage

We adopted the linear spectral unmixing model to map fractional greenspace coverage from the composited Landsat surface reflectance-NDVI-NDWI time series, which can capture subpixel greenspace signals[32]. The linear spectral unmixing model assumes that one pixel's spectral signature (including reflectance and its derivative index) is a linearly weighted sum of a few spectrally pure endmembers and their fraction covers within pixel[68].

$$R_i = \sum_{k=1}^{n} f_{ik} \cdot C_{ik} + \varepsilon_i \tag{1}$$

where $R_i$ represents the spectral signatures of pixel $i$, including spectral reflectance of three visible (i.e., blue, green, red) and one near-

infrared bands, NDVI, and NDWI, $C_{ik}$ represents the spectral signature of the $k$th endmember, $\varepsilon_i$ is the unmodeled residual in the $k$th pixel, $n$ is the total number of endmembers, $f_{ik}$ is the fraction of $k$th endmember within pixel $i$, which is usually calculated from the least-squares method with the following physical constraints:

$$\sum_{k=1}^{n} f_{ik} = 1 \ and \ f_{ik} \geq 0 \ \forall k = 1, \cdots, n \tag{2}$$

We selected vegetation, impervious areas, and water as the three endmembers ($n = 3$). To collect pure spectra of endmembers and minimize their annual variations, in addition to Eq. (2), we included four more physical constraints for three endmembers: (1) vegetation endmembers should have an NDVI value > 0.8, (2) impervious end-members should have an NDVI value < 0.2, (3) water endmembers should have an NDWI value > 0, and (4) three endmembers should be temporally stable over the past two decades, namely, constraints (1–3) should be satisfied for the endmembers for each year from 2000 to 2018.

With the endmember spectra and the associated physical constraints, we first calculated the pixel-level fractional greenspace coverage from Eq. (1). To remove the impacts of residue cloud contaminations, we conducted a pixel-level data smooth to generate the high-quality fractional greenspace coverage time series by using the Savitzky–Golay (SG) filtering approach[69]. We reprojected the 30-m fractional greenspace coverage using the nearest neighbor resampling approach and aggregated it to 100-m resolution to ensure the derived greenspace map is spatially consistent with the 100-m population data. Then, we calculated the city-level physical greenspace coverage rate by overlapping the 100-m resampled pixel-level fractional greenspace coverage with the city boundary and averaging all greenspace coverages of pixels within the city. To further explore the residual uncertainty in the spectral unmixing process, we proposed a spectral unmixing-based threshold classification approach for the sensitivity analysis of physical greenspace coverage mapping. We first generated a greenspace binary map by classifying the 100-m pixel-level fractional greenspace coverage into greenspace (i.e., fractional greenspace coverage ≥ threshold) or non-greenspace (i.e., fractional greenspace coverage < threshold) components using a threshold approach[70]. We adopted the thresholds of 0.3, 0.4, and 0.5 for this classification. Then, we aggregated the greenspace binary map to 100 m for the calculation of city-level physical greenspace coverage.

To validate the accuracy of the Landsat-derived city-level physical greenspace coverage, we used the classification maps of 1-m resolution National Agricultural Imagery Program (NAIP) aerial imagery for 2003–2015 and 10-m resolution Sentinel-2 satellite for 2016-2018 as benchmarks, following the approach in Chen et al. [17]. This task includes three steps. First, we generated the annual composite from Sentinel-2 and NAIP datasets. For Sentinel-2 data, we applied a maximum value composite approach to generate the yearly greenest vegetation green metric across 1028 urban cities like Landsat data. Since NAIP only collected aerial imagery during the agricultural growing season in the sampling regions of the continental United States, we first chose the summer NAIP data (i.e., June–September) of the sampled United States cities as candidates. We then excluded these NAIP candidate data whose observation dates are notably different from the peak growing season by visually comparing them with the corresponding Sentinel-2 and Landsat images. Consequently, a total of 639 United States cities with 19, 580 NAIP images were selected (Supplementary Table 6). Second, we generated the vegetation classification maps from the fine-resolution NAIP and Sentinel-2 imageries using a random forest machine-learning approach. To minimize the impacts of inter-annual variations, we selected training samples of vegetation and non-vegetation pixels from the NAIP and Sentinel-2 imagery for each year to calibrate the random forest algorithm in Google Earth Engine with 15

decision trees while keeping other parameters as default. With the annually calibrated random forest models, we classify the NAIP and Sentinel-2 images into vegetation and non-vegetation binary maps. Third, we calculated city-level greenspace fractions from the binary vegetation maps as references to evaluate the accuracy of Landsat-derived greenspace coverage. The overall consistency results with high Pearson's correlation coefficients (Supplementary Fig. 17) supported the feasibility and acceptable accuracy of using Landsat-derived physical greenspace coverage to measure urban greenspace provision.

### Greenspace exposure

We adopted the population-weighted exposure framework[16–18,32], which can model the spatial interactions between greenspace and population, to quantify the probability and level of human exposure to greenspace according to Eq. (3):

$$GE^d = \frac{\sum_{i=1}^{M} P_i \times G_i^d}{\sum_{i=1}^{M} P_i} \tag{3}$$

where $P_i$ represents the population of pixel $i$, $G_i^d$ represents the fractional greenspace coverage of pixel $i$ that considers both the central and nearby green environment with a buffer size of $d$ (500 m is used in this study), $M$ is the total pixel number within the city, and $GE^d$ is the population-weighted greenspace exposure at a city level.

### Greenspace exposure inequality

We used three commonly used economic metrics (including the Gini coefficient, Atkinson, and Theil indices) to measure the inequality in human exposure to greenspace for global cities following the framework and method of Chen et al. [17]. The calculations of these three metrics are provided in the Supplementary materials. Three metrics range between 0 and 1, where 0 indicates absolute equality and 1 indicates absolute inequality, and increasing value means a larger level of inequality.

### Temporal trend analysis

We used the non-parametric Mann–Kendall statistic and Theil–Sen slope estimator approaches, which are insensitive to data distribution and outliers[71], to calculate the magnitude and direction of the city-level monotonic trends of physical greenspace coverage, human greenspace exposure, and greenspace exposure inequality. We adopted a significance level of 0.05 to assess the significance of time series trends.

### Attribution of changing inequality in greenspace exposure

We proposed a Venn conceptual model to quantify the contributions of greenspace and population to the temporal changes of greenspace exposure inequality. This model originates from the widely used variation partitioning approach that attributes the variations of a response result (i.e., outcome) into different explanatory variables. As shown in Fig. 3, the change of greenspace exposure inequality measured by the Gini index can be decomposed into three parts: individual greenspace provision (region I in Fig. 3a), individual population change (region III in Fig. 3a), and joint greenspace and population impacts (region II in Fig. 3a). When greenspace exposure inequality changes from year $i$ to $i+1$, the individual contribution of greenspace provision (green regions I + II in Fig. 3a) can be modeled as:

$$I + II = \triangle Gini_{green,i} = Gini(g_{i+1}, p_i) - Gini(g_i, p_i) \tag{4}$$

where $\triangle Gini_{green,i}$ denotes the contribution of greenspace provision to the overall changes of greenspace exposure inequality measured by the Gini index in year $i$ ($i = 2001, ..., 2018$), $Gini(g_{i+1}, p_i)$ is the Gini

index with greenspace coverage $g_{i+1}$ in year $i+1$ and population $p_i$ in year $i$, $Gini(g_i, p_i)$ is the Gini index with greenspace coverage $g_i$ in year $i$ and population $p_i$ in year $i$.

Similarly, the individual contribution of population distribution to the overall greenspace exposure inequality (orange regions I + III in Fig. 3a) can be modeled as:

$$II + III = \triangle Gini_{pop,i} = Gini(g_i, p_{i+1}) - Gini(g_i, p_i) \tag{5}$$

where $\triangle Gini_{pop,i}$ denotes the contribution of the population to the overall changes of the Gini index in year $i$ ($i = 2001, ..., 2018$), $Gini(g_i, p_{i+1})$ is the Gini index with greenspace coverage $g_i$ in year $i$ and population $p_{i+1}$ in year $i+1$.

The joint contribution of greenspace provision and population change (regions I + II + III in Fig. 3a) can be modeled as:

$$I + II + III = \triangle Gini_{all,i} = Gini(g_{i+1}, p_{i+1}) - Gini(g_i, p_i) \tag{6}$$

where $\triangle Gini_{all,i}$ denotes the contribution of greenspace provision and population to the overall changes of the Gini index in year $i$ ($i = 2001, ..., 2018$), $Gini(g_{i+1}, p_{i+1})$ is the Gini index with greenspace coverage $g_{i+1}$ and population $p_{i+1}$ in year $i+1$.

By solving Eqs. (4–6), we can quantify the contributions of greenspace provision, population distribution, and joint greenspace and population to the change of greenspace exposure inequality.

In addition to the Venn conceptual model, we adopted another empirical approach[41] to quantify the comparative contributions of greenspace and population to the change of human greenspace exposure inequality, with four major steps: 1) calculation of the monotonic trend of greenspace exposure inequality $\beta_{expo}$ by varying greenspace coverage and population from 2000 to 2018; 2) calculation of greenspace coverage $\beta_{green}$ (or population growth, $\beta_{pop}$) contribution with the monotonic trend analysis by varying greenspace coverage (or population) while keeping a fixed population (or greenspace coverage) at the baseline year 2000; 3) calculation of population growth, $\beta_{pop}$ (or greenspace coverage $\beta_{green}$) contribution by subtracting greenspace (population) contribution from the overall trend of greenspace exposure inequality, i.e., $\beta_{pop} = \beta_{expo} - \beta_{green}$ (or $\beta_{green} = \beta_{expo} - \beta_{pop}$); 4) calculation of the comparative contribution (CC) as: CC = $(|\beta_{pop}| - |\beta_{green}|) \div |\beta_{expo}|$, where $|\cdot|$ represents the absolute function.

## Data availability

Global urban area boundaries products are available from FROM-GLC research group of Tsinghua University (http://data.ess.tsinghua.edu.cn). Greenspace and population datasets used in this study are available from the Google Earth Engine cloud-computing platform. Landsat-5 Collection 2/Tier1 Level-2 surface reflectance product is available at: https://developers.google.com/earth-engine/datasets/catalog/LANDSAT-LT05-C02-T1-L2, Landsat-7 Collection 2/Tier1 Level-2 surface reflectance product is available at: https://developers.google.com/earth-engine/datasets/catalog/LANDSAT-LE07-C02-T1-L2, Landsat-8 Collection 2/Tier1 Level-2 surface reflectance product is available at: https://developers.google.com/earth-engine/datasets/catalog/LANDSAT-LC08-C02-T1-L2, Sentinel-2 Level-2A surface reflectance product is available at: https://developers.google.com/earth-engine/datasets/catalog/COPERNICUS-S2-SR, National Agriculture Imagery Program (NAIP) aerial imagery is available at: https://developers.google.com/earth-engine/datasets/catalog/USDA-NAIP-DOQQ, WorldPop global project population data is available at: https://developers.google.com/earth-engine/datasets/catalog/WorldPop-GP-100m-pop, The resulting greenspace exposure assessments and associated changes for global 1028 cities have been deposited at the following repository: https://datahub.hku.hk/projects/GreenExposureEquality/176019.

## Code availability

The random forest classification algorithm code for greenspace coverage mapping is available from Google Earth Engine: https://developers.google.com/earth-engine/apidocs/ee-classifier-smilerandomforest. The non-parametric Mann-Kendall statistic and Theil–Sen slope estimator codes for temporal trend analysis are available from Google Earth Engine: https://developers.google.com/earth-engine/tutorials/community/nonparametric-trends. The concentrationMetrics library (https://pypi.org/project/concentrationMetrics/) is used to calculate the Gini, Atkinson, and Theil inequality metrics. The code used to produce the necessary data and results in this study is available in the following repository: https://datahub.hku.hk/projects/GreenExposureEquality/176019.

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

## Acknowledgements

This study is supported by The University of Hong Kong HKU-100 Scholars Fund (to B.C.), Seed Fund for Strategic Interdisciplinary Research Scheme Fund (to B.C.), the Research Grants Council of Hong Kong Early Career Scheme (HKU27600222 to B.C.) and General Research Fund (HKU17601423 to B.C.), National Key Research and Development Program of China (2022YFB3903703 to B.X. and B.C.), National Natural Science Foundation of China Young Scientists Fund (42201373 to B.C.), the International Research Center of Big Data for Sustainable Development Goals (CBAS2022GSP04 to P.G., CBAS2022ORP02 to B.X.), the Croucher Foundation (CAS22902/CAS22HU01 to P.G.) and the Major Program of the National Natural Science Foundation of China (42090015 to P.G.).

## Author contributions

B.C. conceived the research idea. B.C. and S.W. designed the study. S.W. and B.C. performed the main data analysis and wrote the manuscript. C.W., B.X., and P.G. reviewed and edited the manuscript.

## Competing interests

The authors declare no competing interests.
