## [Peer Review File · Nature Communications]

Improved human greenspace Exposure equality during 21st century urbanizationEditorial Note: Parts of this Peer Review File have been redacted as indicated to remove third-party material where no permission to publish could be obtained.

REVIEWER COMMENTS

Reviewer #1 (Remarks to the Author):

KEY RESULTS

The key result of the paper is the identification of a different trend in the evolution of inequality in green space exposure depending on the stage of urban development. All cities show a decrease in urban green space exposure inequality but at the origin lie different mechanisms. Cities in the global south have overall a slight decrease in urban green exposure due to rapid urban expansion, while cities in the global north show an increase in urban green exposure.

VALIDITY

The approach taken by the authors is detailed enough to lead to broad conclusions on such a large geographical extent. Nevertheless, I would advise the authors to comment on some of the shortcomings in their analysis and to suggest opportunities for future research in the discussion section.

SIGNIFICANCE

I believe this research is significant in the field since it looks at urban greening and green space exposure inequality from a global perspective. This allows one to take conclusions that transcend local observations.

DATA AND METHODOLOGY

Given the pace of urban development, I believe the Landsat time series is most suitable for an analysis like this. Additionally, spectral unmixing is a suitable method for the assessment of urban green presence because it achieves good results without the need for a location and time-specific reference dataset.

ANALYTICAL APPROACH

The analytical approach in the paper is good however it does not lie in my expertise to give detailed feedback on the quality of the venn conceptual model and the inequality measures that are used in the paper.

SUGGESTED IMPROVEMENTS

Nevertheless, I have some specific remarks/questions for the authors that I believe should be addressed in the paper:

- I would include a section on future research and possible weakness of the analysis. Which nuances of green space provision are being missed in the current analysis?
- Why did you decide to apply a harmonization of the surface reflectance Landsat products rather than using the L2 products?
- Why did you opt for nearest neighbor resampling to map the Landsat green fractions to the population data? Since we are decreasing the resolution I believe that taking the average of the fraction of the pixels that overlap most would give a better idea of the green space fraction in a 100 m tile. (line 552)
- Equation 3 shows how the city-wide green space exposure was calculated but it is unclear how this links to the greenspace exposure parameter that is used for the calculation of the Gini coefficient (g). From reference 18 I understand that the population per tile is linked to a vegetation fraction but I was wondering if the buffer distance to include nearby green environment was also used to calculate this fraction? Or does the fraction only relate to the green coverage in the tile itself?
- For the validation: it is not entirely clear how the validation maps were constructed. In the text, it says that a training set was constructed (with fraction per pixel is assume) from the "annual vegetation green metric". Does this mean that these metrics were used to manually annotate a training set? Or were only "fully green" pixels chosen to train a binary classifier? I believe it is the latter but please reformulate the text to make this clear to the reader.
- In figure 15 in the supplemental material it says that the NAIP and Sentinel-2 imageries are "linear unmixed". I believe this is a mistake?
- Additionally in line 581 it says, "decision tree parameter of 15". Which parameter is this? The number of decision trees in the random forest? What was the depth of the trees, the number of samples per leaf, etc.? It would be interesting to include these parameters in the supplemental

material.

Reviewer #2 (Remarks to the Author):

Recommendation: Major revisions

This paper quantifies the changes in greenness and human exposure to green space in 1028 cities around the world between the years 2000 and 2018. It also analyzes inequality in green space exposure and differences amongst the study cities, noting a difference in changes to exposure inequality between cities in the Global North vs the Global South. Although the paper needs major revisions, once these are addressed, it will offer a timely and novel contribution to the literature and will be of interest to the readers of Nature Communications.

Major Comments:

- The authors do not define what they mean by "greenspace" or what it is they have actually measured in their analysis. They also refer to "greenery" without defining this term. This is problematic because different disciplines understand "greenspace" and "greenery" to mean different things and because this research is measuring a specific type and scale of greenspace as discussed in my next comment.
- I am not convinced that the NDVI threshold of 0.8 is the best choice in this study. A lower threshold might better capture actual greenness exposure, although I realize this comes with a greater risk of classification error. Although the sensitivity analyses show similar trends, they are not the same and the absolute greenspace coverage and exposure levels are markedly different at different NDVI thresholds. The fact that the authors constrained vegetation endmembers to NDVI values above 0.8 is a limitation of the paper and needs to be reflected in the framing of the paper. NDVI values above 0.8 in urban areas typically represent closed-canopy forests such as would be found in a remnant woodland or highly treed park. This dataset does not capture street trees, smaller parks, or most grassy areas. Using lower NDVI thresholds helps to reduce this limitation, but the 30-m resolution of Landsat imagery means that the research is mostly capturing larger green spaces such as parks and woodland areas, rather than distributed greenness such as backyards and street trees. This is not clear in the introduction or general paper framing. In some disciplines, "greenspace" is understood to include distributed greening which is mostly not captured in this analysis.
- Although the authors are careful to use the word "equality" rather than "equity" throughout the paper (with the exception of "equity" in line 45) they need to clearly explain in the introduction and discussion that they are not measuring equity and cannot do so with their methods. For example, although greenspace may be more evenly distributed across a city, this does not mean that those who are most under-resourced have equal (or greater, depending on local definitions of equity) exposure to green space as more privileged populations, or even that under-resourced populations weren't displaced during the creation of green spaces. I appreciate that the authors were careful with their language but the distinction between equity and equality needs to be clearer as Nature Communications is a multi-disciplinary journal and many readers will not understand the distinction. If it is not made clear, readers may come away with the perception that green space is more equitably distributed in cities across the world, when there is no evidence in this paper to support that conclusion.
- On a related note, the authors have not clearly defined "the disparity issue" they introduce in objective 2 (line 100). What is the disparity issue and are they measuring it in this paper?
- Lines 257-263: The implications of this research are overstated here. A global study such as this

cannot identify drivers of change at the scale at which urban planners and developers operate and thus cannot meaningfully inform local-scale urban planning, as suggested by this section. The study provides a helpful analysis of broad trends in urban greening across many cities and its impact should be understood as such. It is not a study that is highly policy-relevant, as most urban greening decisions occur at a municipal or regional scale.

- Where are the limitations of this paper clearly laid out? They need to be clearly included somewhere, ideally in the discussion section.

Minor Comments:

- This paper is well-written but includes grammatical errors throughout. I have noted some below but will leave it to the authors to complete a thorough review to correct errors.

- My preference is to separate "greenspace" into two words ("green space") but this is just a minor suggestion. The authors have used both modes in the manuscript. Please choose one and be consistent.

- Line 28: Change "between" to "from" or change "-" to "and."

- Line 31: Change "contrasting difference" to "contrast."

- Lines 43-44: This sentence needs citations ("This has led to environment injustice, with some communities bearing a disproportionate share of the negative impact of urbanization").

- Line 52: "environment" should be "environmental".

- Lines 68-70: Sentence needs revision for grammar/clarity.

- Lines 87-88: The beginning of this sentence needs citations (i.e. What are these particular cities? Provide some examples via citations).

- Figure 1: Please explain the colour scale more clearly in a and c.

- Line 132: This should be framed as assumed demand rather than demand. Green space is not universally considered to be positive.

- Lines 138-141: This sentence is confusing due to what appear to be grammatical errors. Please revise for clarity.

- Lines 190-191: I suggest softening this sentence. The patterns and effects of urbanization are not consistent around the world.

- Lines 197-198: Sentence needs citation.

- Lines 221-222: The study does not measure why Global North cities have become greener. The authors need to soften this sentence with the word "may" and they need to add citations.

- Lines 222-224: Please provide some examples of these cities.

- Lines 232-233: This does not indicate a benefit of urbanization but simply an increase in greening, perhaps because of and perhaps despite urbanization.

- Methods: I question whether the satellite data should be referred to as high-resolution in this case,

given the scale at which urban areas show green space and other land cover changes. I would prefer to see the resolution specified in the methods rather than the arbitrary "high-resolution" framing, although I acknowledge that others have framed it as such.

- The methodology is sound and enough detail has been provided to reproduce results.

- Lorien Nesbitt

Reviewer #3 (Remarks to the Author):

The manuscript describes a global assessment of urban greenspace coverage, exposure and inequality. Within a time series of almost 20 years, the authors try to observe general trends in the change of coverage of greenspace and its exposure to the urban population. The authors employed a remote sensing methodology to generate a population-weighted greenspace cover in each study area, using Landsat data across the entire time frame. Their results show that there is a general increase in greenspace coverage within the studied cities, and a significant contrast between global north and global south cities. Additionally, significant differences between the greenspace exposure were found between these two groups of cities. This research sheds light on the impact of urbanization on greenspace availability and highlights the need for more equitable distribution of urban nature. However, the manuscript has several flaws that need to be addressed before it can be considered publication worthy. The major issues include the analysis of greenspace inequality drivers in the methods section, the structure of the results section, and the use of ambiguous and confusing terms throughout the manuscript.

Specific comments from the different sections of the manuscript are as follows:

Introduction:

- Line 43: environmental, instead of environment.
- Line 56: Greenspace privilege sounds very much like greenspace access. Why not use a single term for this?
- Authors keep jumping from different concepts related to greenspace, which is confusing. This is seen throughout the introduction section. For example, line 56 and 57 authors talk about greenspace access. And then the succeeding lines talk about greenspace exposure, without linking them. While these concepts are related, they are also distinct from each other, and failing to clearly differentiate between them can make it difficult for readers to follow the authors' arguments.
- Lines 62 to 64: Please expand on this research gap as it is a critical component of the study. Why single year or short time period studies are not sufficient? Do such studies already reveal some trend? Why is it so important to have multiyear assessments? Providing relevant literature to support this framing of the research gap would also be beneficial.
- Line 74: What do you mean by "mature" urban environment?
- Lines 76 to 77: Not sure whether this is always the case. Does this not depend on the type of greenspace too? For example, larger parks or green belts within cities can provide more exposure than smaller green areas.
- Line 80: This conflicts with the statement made in line 68 about it being often chronological.
- Line 87 to 88: Unclear how is this linked to the research gap stated in the following lines.
- Line 99: Would you agree Greenspace stock or Greenspace inventory would be a better term than supply in this context?
- The background information regarding research questions 3 and 4 have not been sufficiently introduced. These two questions could benefit from more explicit and detailed explanation. Please integrate them into the relevant part of introduction section

Methods:

- Line 487: Citation for the GUB product required.

- Line 489: Please simplify the sentence. If I understood correctly, the filtering criteria is the area of cities should be more than 100km²?
- How do you account for the change in urban boundary for each year of the time series?
- Is there a cloud screening criterion as well? For instance, were images with less than 30% cloud cover used?
- Please provide specific information regarding the Landsat images used to generate annual metrics for this study. It would be important to know whether the images were daily, 10-day composites or monthly, as well as the month or season during which the images were used.
- Line 511: Similar to the comment above, the maximum value for within which time period was used (daily, 10-day, monthly etc)?
- Line 542: It seems the NDVI threshold might be too high to capture all types of vegetation for identifying the vegetation end member in the spectral mixed model. Sparse vegetation such as grass or shrubs typically have NDVI values between 0.3 to 0.6, and by setting the threshold at 0.8, the authors may be excluding significant amounts of non-tree vegetation from their analysis. Additionally, this high threshold may also bias the results towards larger and healthier trees, potentially overlooking important information about smaller or less healthy vegetation.
- As a suggestion, it would be helpful for readers to have a visual aid that illustrates the entire mapping process. Therefore, it would be beneficial if the authors could create a flow chart that outlines the steps taken to map the greenspace coverage.
- I have serious reservations about the choice of analysis for investigation the drivers of greenspace inequality. What motivated the authors use this Venn framework for the analysis over other methods? What is the theoretical background behind this? Has this been used in similar studies? If yes, where are the references? If not, why specifically this was used for this dataset. How do you check the robustness of the results from the analysis? There are several open questions unanswered here.

Results and Discussions:

- The structuring of this whole section makes it difficult to read. Please consider reorganizing the whole section. Include subheadings that breakup into smaller, more manageable sections. Additionally, follow a similar flow to the methods section, as this would enhance the readability of the manuscript.
- These sections contain significant amount of ambiguous language, which can make it challenging to follow the authors' arguments. The authors are encouraged to clarify their language to make their points more precise and understandable.
- What is this factor of 0.001?
- Line 117: Does largest temporal trend mean highest magnitude of change during the time period? Please be concise.
- Line 132: Unclear how greenspace supply reflects demand.
- Line 135: Should it not be Fig 1c or 1d?
- A clear explanation of the turning point year of 2011 in the discussion section is missing. The authors should consider providing more context and potential reasons for any significant changes or trends observed in the data around this time.

Response Letter

> We appreciate three reviewers for their time spent assessing this manuscript and for their constructive comments and suggestions, which are very helpful in improving our manuscript.

> Please find our point-by-point responses below to reviewers' specific comments.

Reviewer #1 (Remarks to the Author):

KEY RESULTS

The key result of the paper is the identification of a different trend in the evolution of inequality in green space exposure depending on the stage of urban development. All cities show a decrease in urban green space exposure inequality but at the origin lie different mechanisms. Cities in the global south have overall a slight decrease in urban green exposure due to rapid urban expansion, while cities in the global north show an increase in urban green exposure.

> Thank you very much for the detailed review and summary of our key findings.

VALIDITY

The approach taken by the authors is detailed enough to lead to broad conclusions on such a large geographical extent. Nevertheless, I would advise the authors to comment on some of the shortcomings in their analysis and to suggest opportunities for future research in the discussion section.

> Thanks for the good suggestion. We have added a new paragraph to acknowledge limitations and address prospects from our study in the Discussion of the revised manuscript (Pages 14-16, Lines 303-341), which is duplicated as follows.

"There are also some levels of limitations that should be acknowledged in this study. First, although the integration of 30-m resolution Landsat satellite with a spectral unmixing approach can help resolve the sub-pixel greenspace mapping, our method is still unable to explicitly identify certain greenspace types in small and fragmented patches at the Landsat's resolution, such as street plantation, lawns, and pocket gardens and parks, given the heterogeneous landscape of cities. Therefore, satellites with a higher spatial resolution such as 10-m Sentinel-2, 3-m PlanetScope, and sub-meter WorldView imageries can be incorporated to offer advanced observational opportunities for urban greenspace mapping, but one of the key challenges is how to develop robust spatiotemporal reconstruction algorithms for generating high-quality historical archives. Second, the spatial distributions of greenspace and population footprints within each year of this study are static without modeling spatiotemporal interactions between greenspace and

humans dynamically. On the one hand, people living in cities are spatially moving in their daily routines, being exposed to different nearby green environments beyond those near to their place of residence. On the other hand, seasonal changes of urban greenspace that reflect different phenological phases will influence the greenspace availability over time. Therefore, a promising open topic for follow-up research is to integrate a human mobility dataset with dynamic greenspace observations to enable a spatially and temporally explicit human-greenspace interaction framework and moving towards real-time greenspace exposure assessment. Third, a global sample of 1028 cities are adopted to investigate urban greening and greenspace exposure inequality across a large geographical spectrum, allows us to make conclusions that transcend the local observations that limit many greenspace studies. We acknowledge, however, that the area criterion of 100 km² excludes small and medium-sized cities, home to multiple millions. It will be of interest to compare our results for spatially large cities with a replicated study of smaller cities, since cities of different sizes use space distinctly and have different socio-economic functions and greenspace scales with city size in predictable ways ⁵⁹. These unique processes, in turn, influence the greenspace supply-demand relationship and associated temporal development trajectory, further underscoring the importance in this analysis. One future research direction is to explore the associations of a wider spectrum of urban characteristics (e.g., size, shape, density, function, and socio-economic structure) with greenspace, and measure specific human-centric exposures. Clearly another important factor in assessing the degree and distribution of access or exposure to greenspace is demographic structure such as gender, age, income, and race. In the present study, greenspace exposure differences across population groups are not considered, since their measurement, and a technical consideration of between-group equity is beyond the scope of our methods in this study. We therefore talk in terms of equality, rather than equity of exposure and our measurements relate to human population counts abstracted from specific population sub-groups. A deeper understanding of greenspace exposure and demographic structure such as gender, age, income, and race, is becoming socially and politically urgent ^{18, 60}, which will help identify 'hotspot' areas with less greenspace that disproportionately affect certain vulnerable populations, and potentially help quantify the variations in health outcomes."

References:

- [18] Chen, B., Wu, S., Song, Y., Webster, C., Xu, B., & Gong, P. (2022). Contrasting inequality in human exposure to greenspace between cities of Global North and Global South. *Nature Communications*, 13(1), 4636.
- [59] Fuller, R. A., & Gaston, K. J. (2009). The scaling of green space coverage in European cities. *Biology Letters*, 5(3), 352-355.
- [60] Lu, Y., Chen, L., Liu, X., Yang, Y., Sullivan, W. C., Xu, W., ... & Jiang, B. (2021). Green spaces mitigate racial disparity of health: A higher ratio of green spaces indicates a lower racial disparity in SARS-CoV-2 infection rates in the

USA. *Environment International*, 152, 106465.

SIGNIFICANCE

I believe this research is significant in the field since it looks at urban greening and green space exposure inequality from a global perspective. This allows one to take conclusions that transcend local observations.

> Thank you for your positive comments on the significance of this study.

DATA AND METHODOLOGY

Given the pace of urban development, I believe the Landsat time series is most suitable for an analysis like this. Additionally, spectral unmixing is a suitable method for the assessment of urban green presence because it achieves good results without the need for a location and time-specific reference dataset.

> We appreciate your positive feedback and agreement on the methodology we have employed for urban greenspace mapping, with the integration of Landsat time series and spectral unmixing approach.

ANALYTICAL APPROACH

The analytical approach in the paper is good however it does not lie in my expertise to give detailed feedback on the quality of the venn conceptual model and the inequality measures that are used in the paper.

> Thanks for your good points. We have added more clarifications and additional method comparisons to better elaborate and verify the analytical approaches (i.e., inequality measures and Venn model) we used in this study.

On the one hand, the three inequality measures adopted in this study (i.e., Gini, Atkinson, and Theil), are widely used metrics for quantifying environment exposure inequality, such as urban heat stress (Chakraborty et al. 2019), greenspace (Chen et al. 2022), air pollution (Jbaily et al. 2022), and water scarcity (Ma et al. 2020). In the context of greenspace exposure inequality assessment from this study, the Gini index is calculated by the Lorenz curve framework and quantifies the degree to which the distribution of greenspace exposure deviates from an ideally equal distribution. The Atkinson index is welfare-based inequality metric and measures the proportion of total greenspace in a society that must be foregone to have more equitable distribution of greenspace among individuals in that society. A key attribute of the Atkinson index is its capacity of being decomposed into within and between-group inequality. The Theil index is a generalized entropy measure that quantifies the degree of inequality in the allocation of green spaces over space. A full description on these inequality metrics can be found in Afonso et al. (2015).

The Venn conceptual model originates from the widely used variation partitioning

approach that attributes the variations of a response result (i.e., outcome) into different explanatory variables (Borcard et al. 1992; Delgado-Baquerizo et al. 2020; Legendre et al. 2012; Peres-Neto et al. 2006). For two explanatory variables (X_1 and X_2), as shown in **Fig. R1**, the model can be divided into four components (Borcard et al. 1992; Legendre et al. 2012):

$$[a] = R^2(Y | [X_1, X_2]) - R^2(Y | X_2) \quad (R1)$$

$$[b] = R^2(Y | X_1) + R^2(Y | X_2) - R^2(Y | [X_1, X_2]) \quad (R2)$$

$$[c] = R^2(Y | [X_1, X_2]) - R^2(Y | X_1) \quad (R3)$$

$$[d] = 1 - R^2(Y | [X_1, X_2]) \quad (R4)$$

where the notation ($Y|X$) refers to the analysis (regression or canonical redundancy analysis) of result Y by explanatory variable X and $R^2(Y|X)$ is the variation (R^2) of that analysis.

[redacted]

Fig. R1. The Venn conceptual diagram of variation partitioning results for two explanatory variables X_1 and X_2 . This diagram is revised from Legendre et al. (2012).

Similarly, we used the Venn conceptual model, which has been adopted as a factor separation approach for attributing the drivers of heat exposure (Broadbent et al. 2020), to quantify the individual and joint contributions of greenspace coverage and population distribution in explaining the change of greenspace exposure inequality measured by the Gini index. As shown in **Fig. 3**, we assume the impact from greenspace on the Gini change encompasses two parts: the individual effect (region I, equivalent to **[a]** in **Fig. R1**) and the joint effect of both greenspace and population (region II, equivalent to **[b]** in **Fig. R1**). The total effects of greenspace (C_{I+II}) can be calculated by varying the greenspace coverage while keeping a fixed population.

$$C_{I+II} = \Delta Gini_{green,i} = Gini(g_{i+1}, p_i) - Gini(g_i, p_i) \quad (R5)$$

where g_{i+1} and g_i are greenspace coverage at years $i+1$ and i , respectively. p_i is population at year i . $Gini$ and $\Delta Gini_{green,i}$ is the Gini index and its change induced by greenspace coverage between year i and $i+1$.

Similarly, the total effects of population on the Gini change (C_{II+III}) can be divided into two parts: the individual effect (region III, equivalent to [c] in Fig. R1) and the joint effect of both greenspace and population (region II, equivalent to [b] in Fig. R1). This impact can be calculated by varying the population distribution while keeping a fixed greenspace coverage.

$$C_{II+III} = \Delta Gini_{pop,i} = Gini(g_i, p_{i+1}) - Gini(g_i, p_i) \quad (R6)$$

where p_{i+1} is population at year $i+1$. $\Delta Gini_{pop,i}$ is the Gini change caused by population between year i and $i+1$.

Thus, the total effects of greenspace and population on the Gini change ($C_{I+II+III}$) can be quantified by varying both population distribution and greenspace coverage.

$$C_{I+II+III} = \Delta Gini_{all,i} = Gini(g_{i+1}, p_{i+1}) - Gini(g_i, p_i) \quad (R7)$$

where $\Delta Gini_{all,i}$ is the Gini change caused by both greenspace and population.

By solving Eqs. (R5-7), we can quantify individual and joint contributions from greenspace and population to the Gini change for each year i ($i = 2000, 2001, \dots, 2017, 2018$).

$$C_I = C_{I+II+III} - C_{II+III} \quad (R8)$$

$$C_{III} = C_{I+II+III} - C_{I+II} \quad (R9)$$

$$C_{II} = C_{I+II} + C_{II+III} - C_{I+II+III} \quad (R10)$$

In addition to our Venn conceptual model, we adopted another approach proposed by Tuholske et al. (2020), to empirically quantify the comparative contributions of greenspace and population to the change of greenspace exposure inequality (i.e., the Gini change). This approach assumes the total inequality trend β_{expo} can be linearly decomposed into the contributions of greenspace coverage β_{green} and population distribution β_{pop} (i.e., $\beta_{expo} = \beta_{green} + \beta_{pop}$), the calculation of which, includes four major steps:

(Step-1) Calculate the city-level monotonic trend of greenspace exposure inequality β_{expo} with varying greenspace coverage and population distribution between 2000 and 2018.

(Step-2) Calculate the share of greenspace exposure inequality from greenspace

coverage β_{green} by varying greenspace coverage across year and fixed population at baseline year 2000.

(Step-3) Calculate the share of greenspace exposure inequality trend from population growth β_{pop} by subtracting greenspace contribution from the total trend of greenspace exposure inequality, i.e., $\beta_{pop} = \beta_{expo} - \beta_{green}$. In steps. (2-3), we can also first calculate the share of greenspace exposure inequality from population growth β_{pop} and derive the share from greenspace coverage β_{green} based on the linear model assumption, i.e., $\beta_{green} = \beta_{expo} - \beta_{pop}$.

(Step-4) Normalize the relative contributions as comparative contribution (CC): $CC = (|\beta_{pop}| - |\beta_{green}|) \div |\beta_{expo}|$, where $|\cdot|$ represents the absolute function.

Results show that greenspace dominates the overall change in the Gini index based on the comparative contribution (CC) metric compared to population growth (**Fig. S15**). We have included the description and attribution results of Tuholske et al.'s approach in the revised manuscript, which are duplicated as follows.

(Page 37, Lines 765-775): *"In addition to the Venn conceptual model, we adopted another empirical approach⁴² to quantify the comparative contributions of greenspace and population to the change of human greenspace exposure inequality, with four major steps: 1) calculation of the monotonic trend of greenspace exposure inequality β_{expo} by varying greenspace coverage and population between 2000 and 2018; 2) calculation of greenspace coverage β_{green} (or population growth, β_{pop}) contribution with the monotonic trend analysis by varying greenspace coverage (or population) while keeping a fixed population (or greenspace coverage) at baseline year 2000; 3) calculation of population growth, β_{pop} (or greenspace coverage β_{green}) contribution by subtracting greenspace (population) contribution from the overall trend of greenspace exposure inequality, i.e., $\beta_{pop} = \beta_{expo} - \beta_{green}$ (or $\beta_{green} = \beta_{expo} - \beta_{pop}$); 4) calculation of the comparative contribution (CC) as: $CC = (|\beta_{pop}| - |\beta_{green}|) \div |\beta_{expo}|$, where $|\cdot|$ represents the absolute function."*

(Pages 9-10, Lines 195-200): *"In addition to results based on the Gini index, our sensitivity analysis using the Atkinson and Theil metrics also show very similar attribution results (**Supplementary Figs. 13-14**). We also incorporated the comparative attribution approach⁴² to verify the robustness of the proposed Venn conceptual model, similar results show that the contribution from greenspace dominates the overall change in the Gini index compared to population growth (**Supplementary Fig. S15**)."*

Fig. 3. The Venn conceptual model of quantifying individual and joint contributions from greenspace and population to the overall change in the Gini index. Green circle (regions I and II) denotes the Gini index change $\Delta Gini_{green,i}$ induced by the change of greenspace provision from g_i to g_{i+1} with a fixed population p_i for year i ($i=2001, \dots, 2018$). Orange circle (regions II and III) denotes the Gini index change $\Delta Gini_{pop,i}$ induced by the population change from p_i to p_{i+1} with a fixed greenspace g_i for year i . These two circles (regions I, II, and III) denote the overall change of Gini index $\Delta Gini_{all,i}$ induced by both greenspace and population change from (g_i, p_i) to (g_{i+1}, p_{i+1}) for year i . The overlapped region II denotes the joint effects of greenspace and population changes.

Fig. S15. Comparative contributions of greenspace and population to the temporal change in the Gini index using the empirical approach proposed by Tuholske et al. (2020). **a-b.** Spatial patterns of population versus greenspace for the overall change of the Gini index using the comparative contribution (CC) metric: $CC = (|\beta_{pop}| - |\beta_{green}|) \div |\beta_{expo}|$. **c-d.** City statistics from greenspace and population to the temporal change in greenspace exposure inequality. **a** and **c.** The overall trend β_{expo} and the share of greenspace β_{green} are first calculated and then the share of population is quantified: $\beta_{pop} = \beta_{expo} - \beta_{green}$. **b** and **d.** The overall trend β_{expo} and the share of population β_{pop} are first calculated and then

the share of greenspace is quantified: $\beta_{green} = \beta_{expo} - \beta_{pop}$.

References:

- [1] Afonso, H., LaFleur, M., & Alarcón, D. (2015). Inequality measurement: Development issues no. 2. *Department of Economic and Social Affairs*.
- [2] Borcard, D., Legendre, P., & Drapeau, P. (1992). Partialling out the spatial component of ecological variation. *Ecology*, 73(3), 1045-1055.
- [3] Broadbent, A. M., Krayenhoff, E. S., & Georgescu, M. (2020). The motley drivers of heat and cold exposure in 21st century US cities. *Proceedings of the National Academy of Sciences*, 117(35), 21108-21117.
- [4] Chakraborty, T., Hsu, A., Manya, D., & Sheriff, G. (2019). Disproportionately higher exposure to urban heat in lower-income neighborhoods: a multi-city perspective. *Environmental Research Letters*, 14(10), 105003.
- [5] Chen, B., Wu, S., Song, Y., Webster, C., Xu, B., & Gong, P. (2022). Contrasting inequality in human exposure to greenspace between cities of Global North and Global South. *Nature Communications*, 13(1), 4636.
- [6] Delgado-Baquerizo, M., Reich, P. B., Bardgett, R. D., Eldridge, D. J., Lambers, H., Wardle, D. A., ... & Fierer, N. (2020). The influence of soil age on ecosystem structure and function across biomes. *Nature Communications*, 11(1), 4721.
- [7] Jbaily, A., Zhou, X., Liu, J., Lee, T. H., Kamareddine, L., Verguet, S., & Dominici, F. (2022). Air pollution exposure disparities across US population and income groups. *Nature*, 601(7892), 228-233.
- [8] Legendre, P., Borcard, D., & Roberts, D. W. (2012). Variation partitioning involving orthogonal spatial eigenfunction submodels. *Ecology*, 93(5), 1234-1240.
- [9] Peres-Neto, P. R., Legendre, P., Dray, S., & Borcard, D. (2006). Variation partitioning of species data matrices: estimation and comparison of fractions. *Ecology*, 87(10), 2614-2625..
- [10] Ma, T., Sun, S., Fu, G., Hall, J. W., Ni, Y., He, L., ... & Zhou, C. (2020). Pollution exacerbates China's water scarcity and its regional inequality. *Nature Communications*, 11(1), 650.
- [11] Tuholske, C., Caylor, K., Funk, C., Verdin, A., Sweeney, S., Grace, K., ... & Evans, T. (2021). Global urban population exposure to extreme heat. *Proceedings of the National Academy of Sciences*, 118(41), e2024792118.

SUGGESTED IMPROVEMENTS

Nevertheless, I have some specific remarks/questions for the authors that I believe should be addressed in the paper:

- I would include a section on future research and possible weakness of the analysis. Which nuances of green space provision are being missed in the current analysis?

> Thanks for your good suggestion. Our study used a broad concept of greenspace

coverage (i.e., the fraction of greenspace area within a satellite pixel) to quantify the general pattern of global urban greenspace and the associated indicator of human exposure to greenspace. Although the fine-resolution Landsat footprint (30 x 30 m) and spectral unmixing approaches can help resolve the sub-pixel greenspace mapping issue, our method might miss specific greenspace types in small and fragmented patches, such as street plantation, lawns, and small gardens and parks, given the heterogeneous landscape of cities. Following your suggestion, we have added a new paragraph to acknowledge these limitations and address future prospects from our study in the Discussion section of the revised manuscript (Pages 14-16, Lines 303-341), which is duplicated as follows.

“There are also some levels of limitations that should be acknowledged in this study. First, although the integration of 30-m resolution Landsat satellite with a spectral unmixing approach can help resolve the sub-pixel greenspace mapping, our method is still unable to explicitly identify certain greenspace types in small and fragmented patches at the Landsat’s resolution, such as street plantation, lawns, and pocket gardens and parks, given the heterogeneous landscape of cities. Therefore, satellites with a higher spatial resolution such as 10-m Sentinel-2, 3-m PlanetScope, and sub-meter WorldView imageries can be incorporated to offer advanced observational opportunities for urban greenspace mapping, but one of the key challenges is how to develop robust spatiotemporal reconstruction algorithms for generating high-quality historical archives. Second, the spatial distributions of greenspace and population footprints within each year of this study are static without modeling spatiotemporal interactions between greenspace and humans dynamically. On the one hand, people living in cities are spatially moving in their daily routines, being exposed to different nearby green environments beyond those near to their place of residence. On the other hand, seasonal changes of urban greenspace that reflect different phenological phases will influence the greenspace availability over time. Therefore, a promising open topic for follow-up research is to integrate a human mobility dataset with dynamic greenspace observations to enable a spatially and temporally explicit human-greenspace interaction framework and moving towards real-time greenspace exposure assessment. Third, a global sample of 1028 cities are adopted to investigate urban greening and greenspace exposure inequality across a large geographical spectrum, allows us to make conclusions that transcend the local observations that limit many greenspace studies. We acknowledge, however, that the area criterion of 100 km² excludes small and medium-sized cities, home to multiple millions. It will be of interest to compare our results for spatially large cities with a replicated study of smaller cities, since cities of different sizes use space distinctly and have different socio-economic functions and greenspace scales with city size in predictable ways⁵⁹. These unique processes, in turn, influence the greenspace supply-demand relationship and associated temporal development trajectory, further underscoring the importance in this analysis. One future research direction is to explore the associations of a wider spectrum of urban characteristics

(e.g., size, shape, density, function, and socio-economic structure) with greenspace, and measure specific human-centric exposures. Clearly another important factor in assessing the degree and distribution of access or exposure to greenspace is demographic structure such as gender, age, income, and race. In the present study, greenspace exposure differences across population groups are not considered, since their measurement, and a technical consideration of between-group equity is beyond the scope of our methods in this study. We therefore talk in terms of equality, rather than equity of exposure and our measurements relate to human population counts abstracted from specific population sub-groups. A deeper understanding of greenspace exposure and demographic structure such as gender, age, income, and race, is becoming socially and politically urgent ^{18, 60}, which will help identify ‘hotspot’ areas with less greenspace that disproportionately affect certain vulnerable populations, and potentially help quantify the variations in health outcomes.”

- Why did you decide to apply a harmonization of the surface reflectance Landsat products rather than using the L2 products?

> Thanks for your question. We have already used Landsat Level 2 products in the manuscript, including Landsat-5, Landsat-7, and Landsat-8 Level 2, Collection 2, Tier 1 products. We conducted the surface reflectance harmonization among Landsat-5, Landsat-7, and Landsat-8 sensors following Roy et al.’s (2016) approach to minimize the greenspace estimation bias from spectral characteristic differences across Landsat satellites (<https://developers.google.com/earth-engine/tutorials/community/landsat-etm-to-oli-harmonization>). We have included the data information in the method and data availability sections (Page 30, Lines 616-620), which are duplicated as follows.

“To minimize the uncertainty caused by Landsat-7 scan line off failure ⁶³, we primarily focused on the use of Landsat-5 and Landsat-8 Collection 2/Tier 1 Level-2 products, with data availabilities of 12-year Landsat-5 (2000-2011), 2-year Landsat-7 (2012-2013), and 5-year Landsat-8 (2014-2018).”

“Landsat-5 Collection 2/Tier1 Level-2 surface reflectance product is available at: https://developers.google.com/earth-engine/datasets/catalog/LANDSAT_LT05_C02_T1_L2

Landsat-7 Collection 2/Tier1 Level-2 surface reflectance product is available at: https://developers.google.com/earth-engine/datasets/catalog/LANDSAT_LE07_C02_T1_L2

Landsat-8 Collection 2/Tier1 Level-2 surface reflectance product is available at: https://developers.google.com/earth-engine/datasets/catalog/LANDSAT_LC08_C02_T1_L2”

Reference:

[1] Roy, D. P., Kovalskyy, V., Zhang, H. K., Vermote, E. F., Yan, L., Kumar, S. S., & Egorov, A. (2016). Characterization of Landsat-7 to Landsat-8 reflective wavelength and normalized difference vegetation index continuity. *Remote Sensing of Environment*, 185, 57-70.

- Why did you opt for nearest neighbor resampling to map the Landsat green fractions to the population data? Since we are decreasing the resolution I believe that taking the average of the fraction of the pixels that overlap most would give a better idea of the green space fraction in a 100 m tile. (line 552)

> Thanks for your question. Yes, the nearest neighbor resampling is used for reprojection, and we do use the averaging to reduce the resolution from 30 m to 100 m.

As the Landsat greenspace fractions and WorldPop population datasets come from different sources, two steps are needed to ensure their spatial consistency: (1) data reprojection and (2) spatial resolution aggregation. We first reprojected the Landsat-derived greenspace fractions to the WorldPop population data using the default nearest neighbor resampling approach in Google Earth Engine platform (<https://developers.google.com/earth-engine/guides/resample>) and then aggregated the 30-m reprojected greenspace data to 100-m grid (using the average of greenspace fractions of the pixels) to ensure its spatial consistency with the 100-m WorldPop data set. We have reworded this sentence to avoid confusions in the revised manuscript (Page 32, Lines 667-670), which is duplicated as follows.

“We reprojected the 30-m fractional greenspace coverage using the nearest neighbor resampling approach and aggregated it to 100-m resolution to ensure the derived greenspace map is spatially consistent with the 100-m population data.”

- Equation 3 shows how the city-wide green space exposure was calculated but it is unclear how this links to the greenspace exposure parameter that is used for the calculation of the Gini coefficient (g). From reference 18 I understand that the population per tile is linked to a vegetation fraction but I was wondering if the buffer distance to include nearby green environment was also used to calculate this fraction? Or does the fraction only relate to the green coverage in the tile itself?

> Thanks for your good questions. The Gini index is calculated from the Lorenz curve framework with individual shares of greenspace coverage (**Fig. R2**), rather than directly from the city-level population-weighted greenspace exposure shown in Eq. 3. As shown in **Fig. R2**, Gini is defined as the ratio of the area that lies between the line of equality (i.e., 45-degree line) and the Lorenz curve (i.e., cumulative share of greenspace exposure ranked by residents that are exposed from lowest to highest greenspace) (region A) over the total area under the line

of equality (region *A* plus region *B*) (Chen et al. 2022; Song et al. 2021):

$$Gini = \frac{S_A}{S_A + S_B} \quad (R11)$$

where s_A and s_B represent the areas of regions *A* and *B*, respectively.

Since both x- and y-axis in **Fig. R2b** scale from 0 to 1, we have $S_A + S_B = 0.5$.

Thus, the Gini index can be formulated as:

$$Gini = \frac{S_A}{0.5} = \frac{0.5 - S_B}{0.5} = 1 - 2 \times S_B \quad (R12)$$

The key to solve Eq. (R12) is the area calculation of region *B* that is bounded by the cumulative share of greenspace exposure and the cumulative share of residents from lowest to highest greenspace exposure. To this end, we first used the numerical integration approach to calculate the area of trapezoid B_i that contributed by the i th resident, and then summed each trapezoid area across all residents. The area of trapezoid B_i ($Area_{Bi}$) is calculated as follows:

$$Area_{Bi} = \frac{1}{2} \times \left(\frac{\sum_{j=1}^{i-1} g_j}{\sum_{j=1}^n g_j} + \frac{\sum_{j=1}^i g_j}{\sum_{j=1}^n g_j} \right) \times \frac{1}{n} \quad (R13)$$

where g_j is the greenspace that exposed to the j th resident and n is the total resident number.

Thus, the area of region *B* is calculated as

$$S_B = \sum_{i=1}^n S_{Bi} = \sum_{i=1}^n \frac{1}{2} \times \left(\frac{\sum_{j=1}^{i-1} g_j}{\sum_{j=1}^n g_j} + \frac{\sum_{j=1}^i g_j}{\sum_{j=1}^n g_j} \right) \times \frac{1}{n} \quad (R14)$$

By substituting Eq. (R14) into Eq. (R12), we finally derive the Gini index using the following formula.

$$Gini = 1 - \frac{\sum_{i=1}^n \sum_{j=1}^{i-1} g_j + \sum_{i=1}^n \sum_{j=1}^i g_j}{n \times \sum_{j=1}^n g_j} \quad (R15)$$

Fig. R2. Illustrative diagram of Gini index for inequality assessments of greenspace exposure. The Gini index is defined as the ratio of the area that lies between the line of equality and the Lorenz curve (region A) over the total area under the line of equality (region A plus region B), where Lorenz curve plots the proportion of the greenspace exposure (y-axis) that is cumulatively shared by the residents (x-axis). B_i indicates the contribution of i th residents to the accumulated greenspace exposure and is estimated by the trapezoid area as shown in the left panel, where g_i represents the greenspace that is exposed to the i th resident, and N represents the resident number. Y-axis shows the cumulative share of greenspace exposure; X-axis shows the cumulative share of residents from lowest to highest greenspace exposure.

Yes, we have considered the impacts of nearby green environments on human exposure to greenspace by using a buffer analysis, with the widely used 500-m catchment buffer size. Therefore, the buffered area including nearby green environment will be used to calculate the averaged green fraction for conducting human exposure to greenspace assessment. Moreover, our previous sensitivity analysis in terms of different buffer distances (i.e., 500m, 1km, 1.5km, and 2km) shows that a 500-m buffer distance is appropriate for measuring greenspace exposure and the inequality across global cities (Chen et al. 2022). To clarify this issue, we have included the buffer size information for the calculation of greenspace exposure inequality in the Supplementary information section of the revised manuscript (Page 2, Lines 5-8).

“To consider the impacts of nearby green environments on the inequality measures, we conducted a buffer analysis on greenspace coverage with a size of 500 m using the image convolution algorithm with the “convolve” function in Google Earth Engine.”

References:

- [1] Chen, B., Wu, S., Song, Y., Webster, C., Xu, B., & Gong, P. (2022). Contrasting inequality in human exposure to greenspace between cities of Global North and Global South. *Nature Communications*, 13(1), 4636.
- [2] Song, Y., Chen, B., Ho, H. C., Kwan, M. P., Liu, D., Wang, F., ... & Song, Y. (2021). Observed inequality in urban greenspace exposure in China. *Environment International*, 156, 106778.

- For the validation: it is not entirely clear how the validation maps were constructed. In the text, it says that a training set was constructed (with fraction per pixel is assume) from the “annual vegetation green metric”. Does this mean that these metrics were used to manually annotate a training set? Or were only “fully green” pixels chosen to train a binary classifier? I believe it is the latter but please reformulate the text to make this clear to the reader.

> Thanks for your good questions. We validated the Landsat-derived greenspace coverage mapping using the vegetation and non-vegetation classification maps of 1-m NAIP and 10-m Sentinel-2 imagery. This validation includes three steps. First, we collected the training samples of vegetation and non-vegetation pixels from the fine-resolution NAIP and Sentinel-2 maps for each year to train the random forest algorithm. Second, with the calibrated random forest algorithm, we classified the Sentinel-2 and NAIP images into the vegetation and non-vegetation binary classification maps. Third, the city-level greenspace fractions calculated from the binary classification maps are used as references to validate the accuracy of Landsat-derived greenspace coverage. We have reworded the validation process in the revised manuscript (Pages 33-34, Lines 694-702), which is duplicated as follows.

“Second, we generated the vegetation classification maps from the fine-resolution NAIP and Sentinel-2 imageries using a random forest machine learning approach. To minimize the impacts of inter-annual variations, we selected training samples of vegetation and non-vegetation pixels from the NAIP and Sentinel-2 imagery for each year to calibrate the random forest algorithm in Google Earth Engine with 15 decision trees while keeping other parameters as default. With the annually calibrated random forest models, we classify the NAIP and Sentinel-2 images into vegetation and non-vegetation binary maps. Third, we calculated city-level greenspace fractions from the binary vegetation maps as references to evaluate the accuracy of Landsat-derived greenspace coverage.”

- In figure 15 in the supplemental material is says that the NAIP and Sentinel-2 imageries are “linear unmixed”. I believe this is a mistake?

> Thanks for spotting this typo. We extracted the validation reference of the NAIP and Sentinel-2 by classifying the annual maximum-value composited imageries into vegetation and non-vegetation types, and then calculating the city-level

greenspace fraction for comparison. To make it clear, we have reworded the figure caption of **Supplementary Fig. 17**, which is duplicated as follows.

“Supplementary Fig. 17. Comparison of physical greenspace coverage (GC) derived from 1-m NAIP and 10-m Sentinel-2 imageries (x-axis) and Landsat imagery (y-axis) across global 1028 cities. a. NAIP vs. Landsat in 2003-2015. b-d. Sentinel-2 vs. Landsat in 2016-2018. The GCs from NAIP and Sentinel-2 imageries are extracted from the vegetation and non-vegetation classification mapping with a random forest approach and then aggregated to city-level mean for comparison. Linear regression was used to measure their correlation with Pearson’s r coefficient.”

- Additionally in line 581 it says, “decision tress parameter of 15”. Which parameter is this? The number of decision trees in the random forest? What was the depth of the trees, the number of samples per leaf, etc.? It would be interesting to include these parameters in the supplemental material.

> Thanks for your good questions. Your interpretation is right. The decision tress parameter refers to the number of decision trees in the random forest.

As the classification analysis was conducted on the Google Earth Engine (GEE) cloud-computing platform, we adopted the default random forest function “ee.Classifier.smileRandomForest” to generate the vegetation and non-vegetation classification maps from NAIP and Sentinel-2. This default function provides six parameters (<https://developers.google.com/earth-engine/apidocs/ee-classifier-smilerandomforest>) as follows:

NumberOfTrees: The number of decision trees to create.

VariablesPerSplit: The number of variables per split, default: Null.

MinLeafPopulation: Nodes whose training set contains at least these many points, default: 1.

bagFraction: The fraction of input to bag per tree, default: 0.5.

MaxNodes: The maximum number of leaf nodes in each tree, default: Null.

Seed: The randomization seed, default: 0.

We only changed the NumberOfTrees parameter holding the other parameters at default when using this function. To avoid confusion, we have reworded the use of random forest machine learning approach in the methodology section (Page 33, Lines 696-699), which is duplicated as follows.

“To minimize the impacts of inter-annual variations, we selected training samples of vegetation and non-vegetation pixels from the NAIP and Sentinel-2 imagery for each year to calibrate the random forest algorithm in Google Earth Engine with 15 decision trees while keeping other parameters as default.”

Reviewer #2 (Remarks to the Author):

Recommendation: Major revisions

This paper quantifies the changes in greenness and human exposure to green space in 1028 cities around the world between the years 2000 and 2018. It also analyzes inequality in green space exposure and differences amongst the study cities, noting a difference in changes to exposure inequality between cities in the Global North vs the Global South. Although the paper needs major revisions, once these are addressed, it will offer a timely and novel contribution to the literature and will be of interest to the readers of Nature Communications.

> Thank you for your detailed assessment and positive comments.

Major Comments:

- The authors do not define what they mean by “greenspace” or what it is they have actually measured in their analysis. They also refer to “greenery” without defining this term. This is problematic because different disciplines understand “greenspace” and “greenery” to mean different things and because this research is measuring a specific type and scale of greenspace as discussed in my next comment.

> Thanks for your insightful comment. We agree that greenspace has different definitions for different research communities. Narrowly, it is regarded as outdoor places with significant amounts of vegetation that exist mainly in semi-natural areas and managed parks and gardens and roadside locations (Jim and Chen, 2003). Broadly, greenspace refers to land that is partly or completely covered with grass, trees, shrubs, and other vegetation (Chen et al. 2022a, 2022b; Cameron and Hitchmough, 2016; James et al. 2009). In this study, we used the broad definition of greenspace throughout the manuscript. Following your suggestion, we have replaced the term “greenery” as “greenspace” for a consistent concept.

We also have included our greenspace definition in the methodology section in the revised manuscript (Page 6, Lines 111-113), which is duplicated as follows.

“Specifically, we adopted a broad definition of greenspace to refer to land that is partly or completely covered with grass, trees, shrubs, and other vegetation³³. That is, green-covered land within the curtilage of an urban boundary, since we are focusing on cities in this study.”

References:

- [1] Cameron, R., Hitchmough, J. (2016). Environmental horticulture: science and management of green landscapes. CAB International, Boston: MA.
- [2] Chen, B., Tu, Y., Wu, S., Song, Y., Jin, Y., Webster, C., ... & Gong, P. (2022a).

Beyond green environments: Multi-scale difference in human exposure to greenspace in China. *Environment International*, 166, 107348.

- [3] Chen, B., & Webster, C. (2022b). Eight Reflections on Quantitative Studies of Urban Green Space: A Mapping-Monitoring-Modeling-Management (4M) Perspective. *Landscape Architecture Frontiers* 10(3): 66-77.
- [4] James, P., Tzoulas, K., Adams, M. D., Barber, A., Box, J., Breuste, J., ... & Thompson, C. W. (2009). Towards an integrated understanding of green space in the European built environment. *Urban Forestry & Urban Greening*, 8(2), 65-75.
- [5] Jim, C. Y., & Chen, S. S. (2003). Comprehensive greenspace planning based on landscape ecology principles in compact Nanjing city, China. *Landscape and Urban Planning*, 65(3), 95-116.

• I am not convinced that the NDVI threshold of 0.8 is the best choice in this study. A lower threshold might better capture actual greenness exposure, although I realize this comes with a greater risk of classification error. Although the sensitivity analyses show similar trends, they are not the same and the absolute greenspace coverage and exposure levels are markedly different at different NDVI thresholds. The fact that the authors constrained vegetation endmembers to NDVI values above 0.8 is a limitation of the paper and needs to be reflected in the framing of the paper. NDVI values above 0.8 in urban areas typically represent closed-canopy forests such as would be found in a remnant woodland or highly treed park. This dataset does not capture street trees, smaller parks, or most grassy areas. Using lower NDVI thresholds helps to reduce this limitation, but the 30-m resolution of Landsat imagery means that the research is mostly capturing larger green spaces such as parks and woodland areas, rather than distributed greenness such as backyards and street trees. This is not clear in the introduction or general paper framing. In some disciplines, “greenspace” is understood to include distributed greening which is mostly not captured in this analysis.

> Thanks for your good questions on our methodology regarding greenspace coverage extraction. We employed the widely used linear spectral unmixing model (Haase et al. 2019) to map **sub-pixel greenspace coverage** from Landsat, which assumes that one pixel’s spectral signature is a linearly weighted sum of a few **spectrally pure endmembers** and their **fractional covers** within the target pixel :

$$R_i = \sum_{k=1}^n f_{ik} \cdot C_{ik} + \varepsilon_i \quad (R16)$$

where R_i represents the spectral signatures of pixel i , C_{ik} represents the spectral signature of the k th endmember, ε_i is the unmodeled residual in the i th pixel, n is the total number of endmembers, f_{ik} is the fraction of k th endmember within pixel i . We selected vegetation ($k=1$), impervious area ($k=2$), and water ($k=3$) as three endmembers ($n = 3$) for Eq. (R16) to calculate the vegetation fraction (f_{1i}) as greenspace coverage.

The accuracy of the linear spectral unmixing model highly relies on the purity of the selected endmembers (Dennison and Roberts, 2003; Somers et al. 2011). Considering the lack of spectral library of vegetation endmember, we thus use a high NDVI threshold (i.e., >0.80) to enable the selection of pure vegetation pixels for endmember spectra extraction. According to linear spectral unmixing theory, the NDVI saturates at closed canopy cover, i.e., pixels above high NDVI threshold most likely represent pure vegetation (Fernández-Guisuraga et al. 2020; Schug et al. 2020). To quantify the impacts of NDVI threshold on the result of greenspace coverage extraction, we conducted the sensitivity analysis using different NDVI thresholds of 0.74 - 0.86 with an 0.2 interval holding the other thresholds (impervious area and water) constant for greenspace coverage extraction for 2000-2018. Our results show that the NDVI threshold of 0.80 is very comparable with the other thresholds in extracting city-level absolute greenspace coverage (Fig. R3). Note that the maximum NDVI threshold of 0.86 is used in the sensitivity analysis because of the limited Landsat's spectral response and the constraint of temporally stable endmember over the past two decades (i.e., we require endmember pixels should have a NDVI value over 0.86 for each year within 2000-2018).

Fig. R3. Sensitivity analysis of the NDVI threshold of pure vegetation endmember selection for extracting city-level physical greenspace coverage (GC) across global 1028 cities in 2000-2018. a-f. NDVI threshold of 0.80 vs. a. 0.74. b. 0.76. c. 0.78. d. 0.82. e. 0.84. f. 0.86. Each green circle represents city-level physical GC in a specific year. Linear regression was used

to measure their correlation with Pearson's r coefficient.

As the linear spectral unmixing approach aims to capture sub-pixel greenspace information, our methodology can quantify and incorporate fine-patched greenspace types, such as street trees, shrubs, small parks, and grassy areas (**Fig. R4**).

Fig. R4. City examples of Landsat data for capturing fine-patched greenspace types. a and b. St. Louis city, Missouri, United States. **c and d.** Kansas city, Missouri, United States. **a and c.** 1-m resolution National Agricultural Imagery Program (NAIP) aerial imagery for 2020. **b and d.** 30-m resolution Landsat-derived greenspace for 2020 (USDA Farm Production and Conservation - Business Center, Geospatial Enterprise Operations) Nevertheless, we acknowledge that the 30-m resolution Landsat might miss some small greenspace patches, which require the use of higher-resolution satellite data (e.g., 10-m Sentinel-2 and 3-m PlanetScope imagery). We have discussed our methodology limitation for fine-scale greenspace mapping in the revised manuscript (Pages 14-15, Lines 303-312), which is duplicated as follows.

“There are also some levels of limitations that should be acknowledged in this study. First, although the integration of 30-m resolution Landsat satellite with a spectral unmixing approach can help resolve the sub-pixel greenspace mapping, our method is still unable to explicitly identify certain greenspace types in small and fragmented patches at the Landsat’s resolution, such as street plantation, lawns, and pocket gardens and parks, given the heterogeneous landscape of cities. Therefore, satellites with a higher spatial resolution such as 10-m Sentinel-2, 3-m PlanetScope, and sub-meter WorldView imageries can be incorporated to offer advanced observational opportunities for urban greenspace mapping, but one of

the key challenges is how to develop robust spatiotemporal reconstruction algorithms for generating high-quality historical archives.”

References:

- [1] Dennison, P. E., & Roberts, D. A. (2003). Endmember selection for multiple endmember spectral mixture analysis using endmember average RMSE. *Remote Sensing of Environment*, 87(2-3), 123-135.
- [2] Fernández-Guisuraga, J. M., Calvo, L., & Suárez-Seoane, S. (2020). Comparison of pixel unmixing models in the evaluation of post-fire forest resilience based on temporal series of satellite imagery at moderate and very high spatial resolution. *ISPRS Journal of Photogrammetry and Remote Sensing*, 164, 217-228.
- [3] Haase, D., Jänicke, C., & Wellmann, T. (2019). Front and back yard green analysis with subpixel vegetation fractions from earth observation data in a city. *Landscape and Urban Planning*, 182, 44-54.
- [4] Schug, F., Frantz, D., Okujeni, A., van Der Linden, S., & Hostert, P. (2020). Mapping urban-rural gradients of settlements and vegetation at national scale using Sentinel-2 spectral-temporal metrics and regression-based unmixing with synthetic training data. *Remote Sensing of Environment*, 246, 111810.
- [5] Somers, B., Asner, G. P., Tits, L., & Coppin, P. (2011). Endmember variability in spectral mixture analysis: A review. *Remote Sensing of Environment*, 115(7), 1603-1616.

• Although the authors are careful to use the word “equality” rather than “equity” throughout the paper (with the exception of “equity” in line 45) they need to clearly explain in the introduction and discussion that they are not measuring equity and cannot do so with their methods. For example, although greenspace may be more evenly distributed across a city, this does not mean that those who are most under-resourced have equal (or greater, depending on local definitions of equity) exposure to green space as more privileged populations, or even that under-resourced populations weren’t displaced during the creation of green spaces. I appreciate that the authors were careful with their language but the distinction between equity and equality needs to be clearer as *Nature Communications* is a multi-disciplinary journal and many readers will not understand the distinction. If it is not made clear, readers may come away with the perception that green space is more equitably distributed in cities across the world, when there is no evidence in this paper to support that conclusion.

> Thanks for your insightful comment and good suggestions. Our study is primarily focused on quantifying whether everyone equally shares the same greenspace resource abstracted from income, ethnic, or other group status (Song et al. 2021; Chen et al. 2022; Wu and Kim. 2021), while greenspace equity refers more about the social and environmental justice of the greenspace access, which involves more complex components, such as distributional injustice (i.e., geographic

allocation of greenspace resources), procedural justice (i.e., the equality of the institutional decision-making processes), and interactional injustice (i.e., the interactions that people have on greenspace in an anthropological dimension) (Nesbitt et al. 2018, 2019; Williams et al. 2020).

To avoid confusion, we have replaced the word “equity” in original line 45 with “equality”. We have also defined the scope of measuring greenspace equality in the introduction section (Page 6, Lines 111-113) and discuss our approach for quantifying greenspace equity from the future perspective (Pages 15-16, Lines 332-341) in the revised manuscript, which is duplicated as follows.

(Page 6, Lines 111-113): *“Specifically, we adopted a broad definition of greenspace to refer to land that is partly or completely covered with grass, trees, shrubs, and other vegetation³³. That is, green-covered land within the curtilage of an urban boundary, since we are focusing on cities in this study.”*

(Pages 15-16, Lines 332-341): *“Clearly another important factor in assessing the degree and distribution of access or exposure to greenspace is demographic structure such as gender, age, income, and race. In the present study, greenspace exposure differences across population groups are not considered, since their measurement, and a technical consideration of between-group equity is beyond the scope of our methods in this study. We therefore talk in terms of equality, rather than equity of exposure and our measurements relate to human population counts abstracted from specific population sub-groups. A deeper understanding of greenspace exposure and demographic structure such as gender, age, income, and race, is becoming socially and politically urgent (Chen et al. 2022; Lu et al. 2021), which will help identify ‘hotspot’ areas with less greenspace that disproportionately affect certain vulnerable populations, and potentially help quantify the variations in health outcomes.”*

References:

- [1] Chen, B., Wu, S., Song, Y., Webster, C., Xu, B., & Gong, P. (2022). Contrasting inequality in human exposure to greenspace between cities of Global North and Global South. *Nature Communications*, 13(1), 4636.
- [2] Song, Y., Chen, B., Ho, H. C., Kwan, M. P., Liu, D., Wang, F., ... & Song, Y. (2021). Observed inequality in urban greenspace exposure in China. *Environment International*, 156, 106778.
- [3] Nesbitt, L., Meitner, M. J., Sheppard, S. R., & Girling, C. (2018). The dimensions of urban green equity: A framework for analysis. *Urban Forestry & Urban Greening*, 34, 240-248.
- [4] Nesbitt, L., Meitner, M. J., Girling, C., Sheppard, S. R., & Lu, Y. (2019). Who has access to urban vegetation? A spatial analysis of distributional green equity in 10 US cities. *Landscape and Urban Planning*, 181, 51-79
- [5] Williams, T. G., Logan, T. M., Zuo, C. T., Liberman, K. D., & Guikema, S. D.

(2020). Parks and safety: a comparative study of green space access and inequity in five US cities. *Landscape and Urban Planning*, 201, 103841.

[6] Wu, L., & Kim, S. K. (2021). Exploring the equality of accessing urban green spaces: A comparative study of 341 Chinese cities. *Ecological Indicators*, 121, 107080.

- On a related note, the authors have not clearly defined “the disparity issue” they introduce in objective 2 (line 100). What is the disparity issue and are they measuring it in this paper?

> Thanks for your good question. As we explained above, our target in this study is to measure equality of human exposure to greenspace rather than intergroup equity. The term “disparity issue” is more related to racial, ethnic, and socioeconomic groups. We have replaced this term as “inequality issue” throughout the manuscript, making it clear that the inequality measured is for total urban populations undifferentiated by sub-group characteristics.

- Lines 257-263: The implications of this research are overstated here. A global study such as this cannot identify drivers of change at the scale at which urban planners and developers operate and thus cannot meaningfully inform local-scale urban planning, as suggested by this section. The study provides a helpful analysis of broad trends in urban greening across many cities and its impact should be understood as such. It is not a study that is highly policy-relevant, as most urban greening decisions occur at a municipal or regional scale.

> Thanks for your good comments. This study provides a global-scale pattern of human greenspace exposure and the associated time-series changes that can benefit large-scale urban planning strategies such as greening actions and prioritized practices in a broad, historic, and even forward-looking perspective. Such global awareness might influence the design of local urban planning standards and implementation of municipal or regional greening.

According to your suggestion, we also acknowledged the balance between global vision and local action, and highlighted local-scale practices require multi-dimensional contexts of greening history, greenspace supply status quo, prioritized vulnerable hotspots and the underlying socio-economic factors, and discussed related points in the revised manuscript (Page 14, Lines 290-296), which is duplicated as follows.

“The study tells a broadly positive story of the opening decades of the ‘urban century’, and our analysis of trends, and subsequent studies of positive outliers in those trends, will help cities achieve better net outcomes when planning for balanced changes in urban greenspace loss and construction by incorporating multidimensional contexts of greening history, greenspace supply status quo,

prioritized vulnerable hotspots and the underlying socio-economic factors. The power of big-picture global narrative studies to influence policy-makers, national debates and lobbyists, should also not be underestimated.”

- Where are the limitations of this paper clearly laid out? They need to be clearly included somewhere, ideally in the discussion section.

> Thanks for your good suggestions. We have added a new paragraph to acknowledge limitations and address prospects in the Discussion of the revised manuscript (Pages 14-16, Lines 303-341), which is duplicated as follows.

“There are also some levels of limitations that should be acknowledged in this study. First, although the integration of 30-m resolution Landsat satellite with a spectral unmixing approach can help resolve the sub-pixel greenspace mapping, our method is still unable to explicitly identify certain greenspace types in small and fragmented patches at the Landsat’s resolution, such as street plantation, lawns, and pocket gardens and parks, given the heterogeneous landscape of cities. Therefore, satellites with a higher spatial resolution such as 10-m Sentinel-2, 3-m PlanetScope, and sub-meter WorldView imageries can be incorporated to offer advanced observational opportunities for urban greenspace mapping, but one of the key challenges is how to develop robust spatiotemporal reconstruction algorithms for generating high-quality historical archives. Second, the spatial distributions of greenspace and population footprints within each year of this study are static without modeling spatiotemporal interactions between greenspace and humans dynamically. On the one hand, people living in cities are spatially moving in their daily routines, being exposed to different nearby green environments beyond those near to their place of residence. On the other hand, seasonal changes of urban greenspace that reflect different phenological phases will influence the greenspace availability over time. Therefore, a promising open topic for follow-up research is to integrate a human mobility dataset with dynamic greenspace observations to enable a spatially and temporally explicit human-greenspace interaction framework and moving towards real-time greenspace exposure assessment. Third, a global sample of 1028 cities are adopted to investigate urban greening and greenspace exposure inequality across a large geographical spectrum, allows us to make conclusions that transcend the local observations that limit many greenspace studies. We acknowledge, however, that the area criterion of 100 km² excludes small and medium-sized cities, home to multiple millions. It will be of interest to compare our results for spatially large cities with a replicated study of smaller cities, since cities of different sizes use space distinctly and have different socio-economic functions and greenspace scales with city size in predictable ways (Fuller and Gaston, 2009). These unique processes, in turn, influence the greenspace supply-demand relationship and associated temporal development trajectory, further underscoring the importance in this analysis. One future research direction is to explore the associations of a wider

spectrum of urban characteristics (e.g., size, shape, density, function, and socio-economic structure) with greenspace, and measure specific human-centric exposures. Clearly another important factor in assessing the degree and distribution of access or exposure to greenspace is demographic structure such as gender, age, income, and race. In the present study, greenspace exposure differences across population groups are not considered, since their measurement, and a technical consideration of between-group equity is beyond the scope of our methods in this study. We therefore talk in terms of equality, rather than equity of exposure and our measurements relate to human population counts abstracted from specific population sub-groups. A deeper understanding of greenspace exposure and demographic structure such as gender, age, income, and race, is becoming socially and politically urgent (Chen et al. 2022; Lu et al. 2021), which will help identify ‘hotspot’ areas with less greenspace that disproportionately affect certain vulnerable populations, and potentially help quantify the variations in health outcomes.”

Minor Comments:

- This paper is well-written but includes grammatical errors throughout. I have noted some below but will leave it to the authors to complete a thorough review to correct errors.

> Thanks for spotting the following errors, we have revised accordingly and double-checked the language throughout the manuscript.

- My preference is to separate “greenspace” into two words (“green space”) but this is just a minor suggestion. The authors have used both modes in the manuscript. Please choose one and be consistent.

> Thanks for your good suggestion. We have made it consistent as “greenspace”.

- Line 28: Change “between” to “from” or change “–“ to “and.”

> Revised. Now it reads,

*“Here, we incorporate a Landsat-based 30-meter time-series greenspace mapping and a population-weighted exposure framework to quantify the changes in human exposure to greenspace for 1028 global cities between 2000 **and** 2018.”*

- Line 31: Change “contrasting difference” to “contrast.”

> Revised accordingly (Page 2, Lines 30-33). Now it reads:

*“Nevertheless, we observe a **contrast** in the rate of reduction in greenspace*

exposure inequality between cities in the Global South and North, with a faster rate of reduction in the Global South, nearly four times that of the Global North.”

- Lines 43-44: This sentence needs citations (“This has led to environment injustice, with some communities bearing a disproportionate share of the negative impact of urbanization”).

> Thanks for your good suggestion. We have added the following references to support this sentence (Page 3, Line 42-43). Now it reads:

“This has led to environmental injustice, with some communities bearing a disproportionate share of the negative impact of urbanization ⁸⁻⁹”

References:

- [8] Han, C., Xu, R., Gao, C. X., Yu, W., Zhang, Y., Han, K., ... & Li, S. (2021). Socioeconomic disparity in the association between long-term exposure to PM_{2.5} and mortality in 2640 Chinese counties. *Environment International*, 146, 106241.
- [9] Mitchell, B. C., & Chakraborty, J. (2015). Landscapes of thermal inequity: disproportionate exposure to urban heat in the three largest US cities. *Environmental Research Letters*, 10(11), 115005.

- Line 52: “environment” should be “environmental”.

> Revised accordingly (Page 3, Line 50-51). Now it reads:

*“Greenspace, as the key component of urban nature, offers a wide range of ecosystem services and health benefits, making it an ideal lens for evaluating urban **environmental** sustainability ¹².”*

- Lines 68-70: Sentence needs revision for grammar/clarity.

> We have revised this sentence (Page 4, Line 70-72). Now it reads.

“This replacement often occurs during early-stage urban development via rapid urban expansion, transforming naturally vegetated areas into artificial impervious surfaces and ultimately resulting in a subsequent decrease in greenspace coverage.”

- Lines 87-88: The beginning of this sentence needs citations (i.e. What are these particular cities? Provide some examples via citations).

> Thanks for your good suggestion. We have included the following references to support this sentence. Two example cities are :Shanghai with urbanization-induced

greenspace lose (Wu et al. 2019) and most European cities with urbanization-induced greening (Xu et al. 2022) (Page 5, Line 96-101). Now it reads.

“While both destructive and constructive processes are well understood in the context of particular cities and professional urban management activities³⁹⁻⁴⁰, no study has investigated the net change in greenspace supply and human exposure to greenspace across a global sample of cities for a comparable time, to identify general trends, which may be regarded as net outcomes of the destructive and constructive processes often at work simultaneously in the evolution of city fabric.”

References:

[39] Xu, F., Yan, J., Heremans, S., & Somers, B. (2022). Pan-European urban green space dynamics: A view from space between 1990 and 2015. *Landscape and Urban Planning*, 226, 104477.

[40] Wu, Z., Chen, R., Meadows, M. E., Sengupta, D., & Xu, D. (2019). Changing urban green spaces in Shanghai: Trends, drivers and policy implications. *Land Use Policy*, 87, 104080.

- Figure 1: Please explain the colour scale more clearly in a and c.

> Thanks for your good suggestion. In **Fig. 1a** and **c**, the city-level temporal change trends are divided into four qualitative levels, where cool (dark and slight green) and warm colors (yellow and red) refer to the increasing and decreasing trends of greenspace coverage/exposure, respectively. We have included these explanations into the caption of **Fig. 1**. Now it reads.

“Fig. 1. Temporal changes of physical greenspace coverage (GC) and greenspace exposure (GE) for global 1028 cities from 2000-2018. a. City-level temporal trend of GC changes. b. Annual GC dynamics for Global North and Global South cities. c. City-level temporal trend of GE changes. d. Annual GE dynamics for Global North and Global South cities. In a and c, the city-level temporal change trends are divided into four qualitative levels, where cool (dark and slight green) and warm colors (yellow and red) refer to increasing and decreasing greenspace trends, respectively. The non-parametric Theil–Sen slope estimator approach is used to determine the long-term trends of both GC and GE. The non-parametric Mann-Kendall is used to evaluate the significance of these detected temporal trends. Large bubble sizes represent a statistically significant level of 0.05 (p -value <0.05) and small bubble sizes represent a non-significant trend with p -value >0.05 .”

- Line 132: This should be framed as assumed demand rather than demand. Green space is not universally considered to be positive.

> Thanks. We have replaced the word “demand” with “assumed demand”

accordingly (Page 8, Line 155-157).

*“To adjust gross greenspace supply to reflect **assumed demand**, we used population distribution maps and a population-weighted exposure framework to quantify the temporal change of human greenspace exposure and inequality.”*

- Lines 138-141: This sentence is confusing due to what appear to be grammatical errors. Please revise for clarity.

> We have revised this sentence (Page 8, Line 161-164) and now it reads.

“In North American, European, African, Australian, and Oceanian cities, population distribution amplifies the share of greenspace coverage, with a greater temporal change magnitude in human greenspace exposure compared to physical greenspace coverage.”

- Lines 190-191: I suggest softening this sentence. The patterns and effects of urbanization are not consistent around the world.

> Thanks for your good suggestion. We have rewritten this sentence (Page 11, Lines 218-220) and now it reads.

“Global urbanization might promote increasing population growth and social-economic prosperity but also might enhance environmental degradation, which substantially challenges urban sustainability⁴³⁻⁴⁴.”

- Lines 197-198: Sentence needs citation.

> We have added the following two references to support this sentence (Page 11, Lines 225-226). Now it reads.

“We note, however, this is not entirely true, due to greenspace being a superior economic good, the demand for which rises with income⁴⁶⁻⁴⁷.”

References:

[46] Fruth, E., Kvistad, M., Marshall, J., Pfeifer, L., Rau, L., Sagebiel, J., ... & Winiarski, B. (2019). Economic valuation of street-level urban greening: A case study from an evolving mixed-use area in Berlin. *Land Use Policy*, 89, 104237.

[47] Xiao, Y., Li, Z., & Webster, C. (2016). Estimating the mediating effect of privately-supplied green space on the relationship between urban public green space and property value: Evidence from Shanghai, China. *Land Use Policy*, 54, 439-447.

- Lines 221-222: The study does not measure why Global North cities have become

greener. The authors need to soften this sentence with the word “may” and they need to add citations.

> Thanks for your good suggestion. Following your comments, we have reworded this sentence (Page 12, Lines 249-251) and added the related reference (Anguelovski et al. 2022). Now it reads.

“Global North cities that are highly urbanized have become greener over the past two decades, the likely cause being more green real estate building and green urban planning and city management by both private and public developers and agencies⁵².”

Reference:

[52] Anguelovski, I., Connolly, J. J., Cole, H., Garcia-Lamarca, M., Triguero-Mas, M., Baró, F., ... & Minaya, J. M. (2022). Green gentrification in European and North American cities. *Nature Communications*, 13(1), 3816.

• Lines 222-224: Please provide some examples of these cities.

> Thanks for your good points. As shown in **Fig. 1a**, we can find some cities in North America, Europe, and Russia with a brown color indicating negative trend of greenspace coverage. For clarity, we presented two city examples with different decreasing patterns of greenspace coverage (**Fig. R5**). Greenspace coverage in the Fargo, United States decreased between 2000-2021 and then turned to an increase, with an overall negative trend of greenspace coverage in 2000-2018. Greenspace coverage in Kaliningrad, Russia shows a lasting negative trend in 2000-2018 despite a slight increase between 2000-2003.

Fig. 1. Temporal changes of physical greenspace coverage (GC) and greenspace exposure (GE) for global 1028 cities from 2000-2018. a. City-level temporal trend of GC changes. **b.** Annual GC dynamics for Global North and Global South cities. **c.** City-level temporal trend of GE changes. **d.** Annual GE dynamics for Global North and Global South cities. In **a** and **c**, the city-level temporal change trends are divided into four qualitative levels, where cool (dark and slight green) and warm colors (yellow and red) refer to increasing and decreasing greenspace trends, respectively. The non-parametric Theil–Sen slope estimator approach is used to determine the long-term trends of both GC and GE. The non-parametric Mann-Kendall is used to evaluate the significance of these detected temporal trends. Large bubble sizes represent a statistically significant level of 0.05 (p -value < 0.05) and small bubble sizes represent a non-significant trend with p -value > 0.05 .

Fig. R5. City examples in the Global North experience greenspace loss over the past two decades in **a** Fargo city, North Dakota, United States, and **b** Kaliningrad city, Kaliningrad Oblast, and Russia.

We now have revised this sentence (Pages 12, Line 251-253). Now it reads.

“On the other hand, cities with lower levels of urbanization in the Global North are still experiencing vegetation cover loss (Fig. 1a).”

- Lines 232-233: This does not indicate a benefit of urbanization but simply an increase in greening, perhaps because of and perhaps despite urbanization.

> Thanks for your good suggestion. We have rephrase this sentence (Page 12, Lines 262-263). Now it reads.

“Prior to this, cities globally, were experiencing a decline in greenspace coverage, but after 2011, the trend reversed turning to a positive increase, indicating a net greenspace provisioning.”

- Methods: I question whether the satellite data should be referred to as high-resolution in this case, given the scale at which urban areas show green space and other land cover changes. I would prefer to see the resolution specified in the methods rather than the arbitrary “high-resolution” framing, although I acknowledge that others have framed it as such.

> Thanks for pointing out this issue. We have replaced the term “high-resolution” as “30-m-resolution” in the revised manuscript (Page 29, Lines 593-594). Now it reads.

“We generated two-decade time-series greenspace maps from the 30-m-resolution satellite data for global large cities using the linear spectral unmixing approach.”

- The methodology is sound and enough detail has been provided to reproduce results.

> Thanks for your nice evaluation on our methodology section.

Reviewer #3 (Remarks to the Author):

The manuscript describes a global assessment of urban greenspace coverage, exposure and inequality. Within a time series of almost 20 years, the authors try to observe general trends in the change of coverage of greenspace and its exposure to the urban population. The authors employed a remote sensing methodology to generate a population-weighted greenspace cover in each study area, using Landsat data across the entire time frame. Their results show that there is a general increase in greenspace coverage within the studied cities, and a significant contrast between global north and global south cities. Additionally, significant differences between the greenspace exposure were found between these two groups of cities. This research sheds light on the impact of urbanization on greenspace availability and highlights the need for more equitable distribution of urban nature. However, the manuscript has several flaws that need to be addressed before it can be considered publication worthy. The major issues include the analysis of greenspace inequality drivers in the methods section, the structure of the results section, and the use of ambiguous and confusing terms throughout the manuscript.

> Thanks for your detailed assessment and comments. According to your suggestions, we have carefully revised our manuscript, including the explanation of Venn-based attribution model (i.e., variation partitioning approach), the reorganization of result section, and the removal of confusing expressions and terms.

Specific comments from the different sections of the manuscript are as follows:

> Thank you. Please see our point-by-point responses below.

Introduction:

- Line 43: environmental, instead of environment.

> Thanks, we revised it accordingly (Page 3, Lines 42-43). Now it reads.

“This has led to environmental injustice, with some communities bearing a disproportionate share of the negative impact of urbanization⁸⁻⁹”

- Line 56: Greenspace privilege sounds very much like greenspace access. Why not use a single term for this?

> Thanks for this good suggestion. We here used the term “Greenspace privilege” to represent human exposure to greenspace. We have rewritten this sentence (Page 3, Lines 55-57). Now it reads.

“Human exposure to greenspace, measured as the averaged amount of greenspace coverage within people’s nearby environment expressed as a percentage¹⁸, has also been quantified in sampled cities¹⁹⁻²⁰”

- Authors keep jumping from different concepts related to greenspace, which is confusing. This is seen throughout the introduction section. For example, line 56 and 57 authors talk about greenspace access. And then the succeeding lines talk about greenspace exposure, without linking them. While these concepts are related, they are also distinct from each other, and failing to clearly differentiate between them can make it difficult for readers to follow the authors' arguments.

> Thanks for pointing out this issue. Greenspace access typically estimates proximity to urban greenspace. Exposure metrics derive from remote sensing imagery, such as the Normalized Difference Vegetation Index (NDVI), measures surrounding greenness. Unlike accessibility that usually expressed as a distance-value, exposure is generally quantified as the proportional value of greenspace within a specified area, for example, a buffer area around a residence used in this study (Jarvis et al. 2020; Reid et al. 2018).

To avoid confusion, we used the term “greenspace exposure” throughout the manuscript to indicate the possibility of human sharing greenspace.

References:

- [1] Jarvis, I., Gergel, S., Koehoorn, M., & van den Bosch, M. (2020). Greenspace access does not correspond to nature exposure: Measures of urban natural space with implications for health research. *Landscape and Urban Planning*, 194, 103686.
- [2] Reid, C. E., Kubzansky, L. D., Li, J., Shmool, J. L., & Clougherty, J. E. (2018). It's not easy assessing greenness: a comparison of NDVI datasets and neighborhood types and their associations with self-rated health in New York City. *Health & Place*, 54, 92-101.

- Lines 62 to 64: Please expand on this research gap as it is a critical component of the study. Why single year or short time period studies are not sufficient? Do such studies already reveal some trend? Why is it so important to have multiyear assessments? Providing relevant literature to support this framing of the research gap would also be beneficial.

> Thanks for your good questions and suggestions. The temporal impacts of urbanization on greenspace are twofold. First, the urban sprawl or expansion process that might cause green space loss through land cover conversion is a long-term phenomenon (Chen et al. 2020; Gao and Neill, 2020). Second, the positive effects of urbanization on greenspace through the effects of fertilization, irrigation, warmer temperatures, greater tropospheric CO₂ concentrations, higher

atmospheric nitrogen deposition, as well as greenspace management activities are relatively slow temporal evolution processes (Du et al. 2019; Mattijssen et al. 2017). The single-year or short time period (e.g., 5 years) studies reflect greenspace patterns over baseline year (Chen et al. 2022; He et al. 2023) but cannot fully capture the temporal details of urbanization's impacts on greenspace and attribute the associated causality.

We have rephased this sentence (Page 4, Lines 62-66). Now it reads.

“However, existing studies on greenspace exposure and inequality are constrained to data from individual baseline years that cannot capture long-term urbanization process²¹⁻²² and gradual temporal evolution of human-nature interactions such as urban warming and green-space management strategy impacts²³⁻²⁴, limiting our understanding of the impact of urban development on greenspace supply, human exposure, and inequality over time.”

References:

- [1] Chen, B., Wu, S., Song, Y., Webster, C., Xu, B., & Gong, P. (2022). Contrasting inequality in human exposure to greenspace between cities of Global North and Global South. *Nature Communications*, 13(1), 4636.
- [2] Chen, G., Li, X., Liu, X., Chen, Y., Liang, X., Leng, J., ... & Huang, K. (2020). Global projections of future urban land expansion under shared socioeconomic pathways. *Nature Communications*, 11(1), 537.
- [3] Du, J., Fu, Q., Fang, S., Wu, J., He, P., & Quan, Z. (2019). Effects of rapid urbanization on vegetation cover in the metropolises of China over the last four decades. *Ecological Indicators*, 107, 105458.
- [4] Gao, J., & O'Neill, B. C. (2020). Mapping global urban land for the 21st century with data-driven simulations and Shared Socioeconomic Pathways. *Nature Communications*, 11(1), 2302.
- [5] Han, Y., He, J., Liu, D., Zhao, H., & Huang, J. (2023). Inequality in urban green provision: A comparative study of large cities throughout the world. *Sustainable Cities and Society*, 89, 104229.
- [6] Mattijssen, T. J. M., Van der Jagt, A. P. N., Buijs, A. E., Elands, B. H. M., Erlwein, S., & Laforteza, R. (2017). The long-term prospects of citizens managing urban green space: From place making to place-keeping?. *Urban Forestry & Urban Greening*, 26, 78-84.

- Line 74: What do you mean by “mature” urban environment?

> Thanks. We here refer to well-developed urban environment. Revised accordingly (Pages 4-5, Lines 80-83). Now it reads.

“City management practices, such as planting street trees and creating vertical gardens, can also increase the overall greenspace supply in an urban environment

developed over time under a legally strong environmental planning regime ³¹⁻³²”

- Lines 76 to 77: Not sure whether this is always the case. Does this not depend on the type of greenspace too? For example, larger parks or green belts within cities can provide more exposure than smaller green areas.

> Thanks for this good question. Yes, different green landscapes regarding types, sizes, and compositions will demonstrate distinct green exposure values, underlying the ‘luxury effects’. Wealthier neighborhoods can invest more in the maintenance and management efforts of urban greenspace, which in turn increases greenspace supply, species, planting diversity, and aesthetic landscape qualities (Avolio et al. 2018; Leong et al. 2018; Zhang et al. 2022).

According to your suggestion, we have revised this sentence (Page 5, Lines 85-88), which is duplicated as below.

“As well as a crowding effect of population growth, leading to reduced greenspace exposure, an income effect tends to increase greenspace supply, diversity, and aesthetics in wealthier cities, since greenspace is a ‘superior good’, with demand increasing as the prosperity of citizens increases ³⁴.”

References:

- [1] Avolio, M. L., Pataki, D. E., Trammell, T. L., & Endter-Wada, J. (2018). Biodiverse cities: the nursery industry, homeowners, and neighborhood differences drive urban tree composition. *Ecological Monographs*, 88(2), 259-276.
- [2] Leong, M., Dunn, R. R., & Trautwein, M. D. (2018). Biodiversity and socioeconomics in the city: a review of the luxury effect. *Biology Letters*, 14(5), 20180082.
- [3] Zhang, H. L., Cubino, J. P., Nizamani, M. M., Harris, A. J., Cheng, X. L., Da, L., ... & Wang, H. F. (2022). Wealth and land use drive the distribution of urban green space in the tropical coastal city of Haikou, China. *Urban Forestry & Urban Greening*, 71, 127554.

- Line 80: This conflicts with the statement made in line 68 about it being often chronological.

> Thanks for pointing out this issue. We have revised this sentence on the greenspace loss in early-stage urban development (Page 4, Lines 70-78). Now it reads.

“This replacement often occurs during early-stage urban development via rapid urban expansion, transforming naturally vegetated areas into artificial impervious surfaces and ultimately resulting in a subsequent decrease in greenspace coverage. Historically, there is typically a process of greenspace destruction as

agricultural and natural green land is replaced by impervious surfaces and then a later process of re-greening via environmental improvements. Urban expansion through comprehensive spatial planning (as opposed to incremental unplanned sprawl), tends to 'reprovision' greenspace by design from the outset, but these spaces themselves tend to evolve (decrease and increase in quantity and quality) over time according to income, land value and environmental preferences²⁷⁻²⁸."

- Line 87 to 88: Unclear how is this linked to the research gap stated in the following lines.

> Thanks for your good comment. The destructive and constructive processes impact the net change of greenspace supply and human exposure. We have revised this sentence (Page 5, Line 96-101). Now it reads.

"While both destructive and constructive processes are well understood in the context of particular cities and professional urban management activities³⁹⁻⁴⁰, no study has investigated the net change in greenspace supply and human exposure to greenspace across a global sample of cities for a comparable time, to identify general trends, which may be regarded as net outcomes of the destructive and constructive processes often at work simultaneously in the evolution of city fabric."

- Line 99: Would you agree Greenspace stock or Greenspace inventory would be a better term than supply in this context?

> Thanks for your insightful suggestion. Greenspace stock refers to the amount of green space that exists within urban areas and can be measured in terms of the total area of green space per capita or as a percentage of the total land area of the urban area (Huang et al. 2021a, 2021b). Greenspace inventory refers to the characteristics and features of the urban green space, such as the types of vegetation, the age and condition of trees, the presence of wetlands or streams, as well as the availability of amenities like benches, playgrounds, or walking paths (Feltynowski et al. 2018; Vogt and Fischer, 2017). Compared to greenspace stock and greenspace inventory, greenspace supply has more implications for urban greenspace planning because it reflects more about the greenspace provision and human needs in a supply-demand perspective (Boulton et al. 2018; Liu et al. 2020).

References:

- [1] Boulton, C., Dedekorkut-Howes, A., & Byrne, J. (2018). Factors shaping urban greenspace provision: A systematic review of the literature. *Landscape and Urban Planning*, 178, 82-101.
- [2] Feltynowski, M., Kronenberg, J., Bergier, T., Kabisch, N., Łaszkiwicz, E., & Strohbach, M. W. (2018). Challenges of urban green space management in the face of using inadequate data. *Urban Forestry & Urban Greening*, 31, 56-66.

- [3] Huang, Y., Lin, T., Zhang, G., Zhu, Y., Zeng, Z., & Ye, H. (2021). Spatial patterns of urban green space and its actual utilization status in China based on big data analysis. *Big Earth Data*, 5(3), 391-409.
- [4] Huang, Y., Lin, T., Xue, X., Zhang, G., Liu, Y., Zeng, Z., ... & Sui, J. (2021). Spatial patterns and inequity of urban green space supply in China. *Ecological Indicators*, 132, 108275.
- [5] Liu, H., Remme, R. P., Hamel, P., Nong, H., & Ren, H. (2020). Supply and demand assessment of urban recreation service and its implication for greenspace planning-A case study on Guangzhou. *Landscape and Urban Planning*, 203, 103898.
- [6] Vogt, J. M., & Fischer, B. C. (2017). A protocol for citizen science monitoring of recently-planted urban trees. *Urban Forests, Ecosystem Services and Management*; Blum, J., Ed, 153-186.

- The background information regarding research questions 3 and 4 have not been sufficiently introduced. These two questions could benefit from more explicit and detailed explanation. Please integrate them into the relevant part of introduction section

> Thanks for your good comments. Answering questions 3 and 4 can help us learn the spatiotemporal patterns and the drivers of greenspace and then improve our understanding of future projection of urban greenspace. We have included the background information on these two questions in the revised manuscript (Pages 5-6, Lines 101-107), which is duplicated as follows.

“Inequality in greenspace exposure is an increasing concern as it can be translated into adverse effects in mental and physical health¹⁹⁻²⁰. Individual studies show how greenspace provision and the joint effects of greenspace provision and spatial configuration control greenspace exposure inequality¹⁸. However, the drivers of changing greenspace exposure inequality over time remain elusive and limit our understanding of the relationship between greenspace and population distributions in determining and attempting to promote greenspace exposure equality in the future projection.”

Methods:

- Line 487: Citation for the GUB product required.

> Thanks for pointing out this issue. We have included the reference (Li et al. 2020) (Page 29, Lines 603-604). Now it reads:

“The boundaries of these urban areas were extracted from the latest global urban boundaries (GUB) product⁵⁹.”

Reference:

[59] Li, X., Gong, P., Zhou, Y., Wang, J., Bai, Y., Chen, B., ... & Zhu, Z. (2020). Mapping global urban boundaries from the global artificial impervious area (GAIA) data. *Environmental Research Letters*, 15(9), 094044.

- Line 489: Please simplify the sentence. If I understood correctly, the filtering criteria is the area of cities should be more than 100km²?

> Thanks. Yes, we have now revised this sentence (Page 29, Lines 604-608). Now it reads.

“As the high-quality year-by-year GUB datasets are lacking, we chose the GUB data in 2018 with a geographic area large than 100 km² as urban boundary to allow for: 1) the exploration of the urban expansion impacts on greenness change and 2) the collection of sufficient samples for measuring human greenspace exposure and inequality¹⁸.”

- How do you account for the change in urban boundary for each year of the time series?

> Thanks for your good question. We used the global urban boundaries product for 2018 (i.e., GUB_2018) to extract greenspace time series of urban extent for two reasons. On one hand, as urban population is primarily resided in urban areas (**Fig. R6a**), the differences between city-level greenspace exposure and the associated inequality extracted using the actual urban boundary and the bigger GUB_2018 used in this study are minor. Using the 5-year interval global urban boundaries product for 2005 and 2010, comparison results show close relationships between greenspace exposure and Gini levels from actual urban boundaries (i.e., GUB_2005 and 2010) and GUB_2018 for year 2005 and 2010 (**Fig. R6b-e**). On the other hand, the high-quality year-by-year urban boundary product with temporal consistency is still lacking for global urban greenspace analysis.

We explained this reason in the revised manuscript (Page 29, Lines 604-608), which is duplicated as follows.

“As the high-quality year-by-year GUB datasets are lacking, we chose the GUB data in 2018 with a geographic area large than 100 km² as urban boundary to allow for: 1) the exploration of the urban expansion impacts on greenness change and 2) the collection of sufficient samples for measuring human greenspace exposure and inequality¹⁸.”

Fig. R6. Minor difference from the dynamic urban boundary for the city-level greenspace exposure estimation compared to the global urban boundary product for 2018. a. Scheme diagram of the difference induced from different urban boundaries (true urban boundary vs. GUB_2018 used in this study) for modeling human-greenspace interactions. **b-e.** Close relationships between city-level greenspace exposure and inequality extracted from different urban boundaries. **b** and **c** refer to GUB_2005 vs. GUB_2018. **c** and **e** refer to GUB_2010 vs. GUB_2018.

- Is there a cloud screening criterion as well? For instance, were images with less than 30% cloud cover used?

> Thanks for your question. We used the pixel-level quality assurance (QA) layer data (i.e., the QA_PIXEL band in the Landsat surface reflectance product as shown in https://developers.google.com/earth-engine/datasets/catalog/LANDSAT_LC08_C02_T1_L2#bands) instead of image-level cloud screening to remove pixels that are contaminated by clouds, cloud shadow, and snow. Moreover, we used the maximum value composite approach to generate the annual greenest vegetation green metrics, which can further minimize the cloud cover impacts. We have specified these two points in the original manuscript (Page 30, Lines 624-630), which are duplicated as follows.

“With the QA bitmask layer, we excluded the pixels that were contaminated by cloud cover, shadow, and snow. We further calculated the normalized difference vegetation index (NDVI) and normalized difference water index (NDWI) to quantify

the spectral characteristics of green vegetation and water body, respectively. Finally, we adopted a maximum value composite approach to generate the annual greenest vegetation green metrics by selecting the pixel-based maximum NDVI values from the cloud-free time series within a one-year cycle.”

- Please provide specific information regarding the Landsat images used to generate annual metrics for this study. It would be important to know whether the images were daily, 10-day composites or monthly, as well as the month or season during which the images were used.

> Thanks for your question. We generated the annual greenspace metrics from **the surface reflectance products of three visible (i.e., blue, green, and red), one near-infrared bands, the associated normalized difference vegetation index (NDVI) and normalized difference water index (NDWI)** with three steps: (1) cloud removal using the pixel-level quality assurance (QA) layer data, (2) annual greenspace metric composition using the maximum value composite approach; (3) greenspace coverage extraction using the linear spectral unmixing method. As Landsat satellites have a 16-day temporal resolution, repeatedly observing the same place on Earth with a 16-day revisit cycle, we composed Landsat satellite images in a one-year cycle as the annual greenspace metric. We have clarified these points in the revised manuscript (Page 30, Lines 627-630), which is duplicated as follows.

“Finally, we adopted a maximum value composite approach to generate the annual greenest vegetation green metrics by selecting the pixel-based maximum NDVI values from the cloud-free time series within a one-year cycle.”

- Line 511: Similar to the comment above, the maximum value for within which time period was used (daily, 10-day, monthly etc)?

> Thanks for your question. We used the pixel-level maximum NDVI value within a one-year cycle as the greenspace metric. We have rewritten this sentence for clarity (Page 30, Lines 627-630), which is duplicated as follows.

“Finally, we adopted a maximum value composite approach to generate the annual greenest vegetation green metrics by selecting the pixel-based maximum NDVI values from the cloud-free time series within a one-year cycle.”

- Line 542: It seems the NDVI threshold might be too high to capture all types of vegetation for identifying the vegetation end member in the spectral mixed model. Sparse vegetation such as grass or shrubs typically have NDVI values between 0.3 to 0.6, and by setting the threshold at 0.8, the authors may be excluding significant amounts of non-tree vegetation from their analysis. Additionally, this high threshold may also bias the results towards larger and healthier trees, potentially

overlooking important information about smaller or less healthy vegetation.

> Thanks for your good question. We employed the widely used linear spectral unmixing model (Haase et al. 2019) to map **sub-pixel greenspace coverage** from Landsat, which assumes that one pixel's spectral signature is a linearly weighted sum of a few **spectrally pure endmembers** and their **fractional covers** within the target pixel :

$$R_i = \sum_{k=1}^n f_{ik} \cdot C_{ik} + \varepsilon_i \quad (\text{R17})$$

where R_i represents the spectral signatures of pixel i , C_{ik} represents the spectral signature of the k th endmember, ε_i is the unmodeled residual in the i th pixel, n is the total number of endmembers, f_{ik} is the fraction of k th endmember within pixel i . We selected vegetation ($k=1$), impervious area ($k=2$), and water ($k=3$) as three endmembers ($n = 3$) for Eq. (R17) to calculate the vegetation fraction (f_{i1}) as greenspace coverage.

The accuracy of the linear spectral unmixing model relies on the purity of the selected endmembers (Dennison and Roberts, 2003; Somers et al. 2011). Considering the lack of spectral library of vegetation endmember, we thus use a high NDVI threshold (i.e., >0.80) to enable the selection of pure vegetation pixels for endmember spectra extraction. According to linear spectral unmixing theory, the NDVI saturates at closed canopy cover, i.e., pixels above high NDVI threshold most likely represent pure vegetation (Fernández-Guisuraga et al. 2020; Schug et al. 2020). To quantify the impacts of NDVI threshold on the result of greenspace coverage extraction, we conducted a sensitivity analysis using different NDVI thresholds of 0.74-0.86 with an 0.2 interval holding the other thresholds (impervious area and water) constant for greenspace coverage extraction for 2000-2018. Our results show that the NDVI threshold of 0.80 is very comparable with the other thresholds in extracting city-level absolute greenspace coverage (**Fig. R6**). Note that the maximum NDVI threshold of 0.86 is used here because of the limited Landsat's spectral response and the constraint of temporally stable endmember over the past two decades (i.e., we require endmember pixels should have a NDVI value over 0.86 for each year within 2000-2018).

Fig. R6. Sensitivity analysis of the NDVI threshold of pure vegetation endmember selection for extracting city-level physical greenspace coverage (GC) across global 1028 cities in 2000-2018. a-f. NDVI threshold of 0.80 vs. **a** 0.74. **b**. 0.76. **c**. 0.78. **d**. 0.82. **e**. 0.84. **f**.0.86. Each green circle represents city-level physical GC in a specific year. Linear regression was used to measure their correlation with Pearson's r coefficient.

As the linear spectral unmixing approach calculates sub-pixel greenspace information, it to some extent can detect fine-patched greenspace types, such as street trees, shrubs, small parks, and grassy areas. Nevertheless, we acknowledge that the 30-m resolution Landsat might miss some small greenspace patches, which require the use of higher-resolution satellite data (e.g., 10-m Sentinel-2 and 3-m PlanetScope imagery). We have discussed our methodology limitation for fine-scale greenspace mapping in the revised manuscript (Pages 14-15, Lines 303-312), which is duplicated as follows.

“There are also some levels of limitations that should be acknowledged in this study. First, although the integration of 30-m resolution Landsat satellite with a spectral unmixing approach can help resolve the sub-pixel greenspace mapping, our method is still unable to explicitly identify certain greenspace types in small and fragmented patches at the Landsat’s resolution, such as street plantation, lawns, and pocket gardens and parks, given the heterogeneous landscape of cities. Therefore, satellites with a higher spatial resolution such as 10-m Sentinel-2, 3-m

PlanetScope, and sub-meter WorldView imageries can be incorporated to offer advanced observational opportunities for urban greenspace mapping, but one of the key challenges is how to develop robust spatiotemporal reconstruction algorithms for generating high-quality historical archives.”

References:

- [1] Dennison, P. E., & Roberts, D. A. (2003). Endmember selection for multiple endmember spectral mixture analysis using endmember average RMSE. *Remote Sensing of Environment*, 87(2-3), 123-135.
- [2] Fernández-Guisuraga, J. M., Calvo, L., & Suárez-Seoane, S. (2020). Comparison of pixel unmixing models in the evaluation of post-fire forest resilience based on temporal series of satellite imagery at moderate and very high spatial resolution. *ISPRS Journal of Photogrammetry and Remote Sensing*, 164, 217-228.
- [3] Haase, D., Jänicke, C., & Wellmann, T. (2019). Front and back yard green analysis with subpixel vegetation fractions from earth observation data in a city. *Landscape and Urban Planning*, 182, 44-54.
- [4] Schug, F., Frantz, D., Okujeni, A., van Der Linden, S., & Hostert, P. (2020). Mapping urban-rural gradients of settlements and vegetation at national scale using Sentinel-2 spectral-temporal metrics and regression-based unmixing with synthetic training data. *Remote Sensing of Environment*, 246, 111810.
- [5] Somers, B., Asner, G. P., Tits, L., & Coppin, P. (2011). Endmember variability in spectral mixture analysis: A review. *Remote Sensing of Environment*, 115(7), 1603-1616.

- As a suggestion, it would be helpful for readers to have a visual aid that illustrates the entire mapping process. Therefore, it would be beneficial if the authors could create a flow chart that outlines the steps taken to map the greenspace coverage.

> Thanks for your good question. We have added the flowchart outlining the entire research design in the method summary section (Page 29, Lines 599-600), which is shown as follows.

Fig. S16. Flowchart of the research design in this study with five major steps. **a.** Data preparations of global city boundary and population datasets. **b.** Greenspace mapping from long-term Landsat satellite imagery with the linear spectral unmixing approach. **c.** human exposure to greenspace with population-weighted exposure model and inequality assessment. **d.** spatiotemporal analysis

of physical greenspace coverage, human exposure, and inequality over global 1028 cities. **e.** Driver attribution of temporal change in greenspace exposure inequality over past two decades by accounting for individual greenspace and population effects, and their interaction effects.

- I have serious reservations about the choice of analysis for investigation the drivers of greenspace inequality. What motivated the authors use this Venn framework for the analysis over other methods? What is the theoretical background behind this? Has this been used in similar studies? If yes, where are the references? If not, why specifically this was used for this dataset. How do you check the robustness of the results from the analysis? There are several open questions unanswered here.

> Thanks for your good question. Venn diagram model originates from the widely used variation partitioning approach that attributes the variations of a response result into different explanatory variables (Borcard et al. 1992; Delgado-Baquerizo et al. 2020; Legendre et al. 2012; Peres-Neto et al. 2006). For two explanatory variables (X_1 and X_2), the model can be divided into four components (**Fig. R7**; Borcard et al. 1992; Legendre et al. 2012):

$$[a] = R^2(Y | [X_1, X_2]) - R^2(Y|X_2) \quad (R18)$$

$$[b] = R^2(Y|X_1) + R^2(Y|X_2) - R^2(Y | [X_1, X_2]) \quad (R19)$$

$$[c] = R^2(Y | [X_1, X_2]) - R^2(Y|X_1) \quad (R20)$$

$$[d] = 1 - R^2(Y | [X_1, X_2]) \quad (R21)$$

where the notation ($Y|X$) refers to the analysis (regression or canonical redundancy analysis) of result Y by explanatory variable X and $R^2(Y|X)$ is the variation (R^2) of that analysis.

[redacted]

Fig. R7. The Venn conceptual diagram of variation partitioning results for two explanatory variables X_1 and X_2 . This diagram is revised from Legendre et al.

(2012).

Similarly, we used the Venn conceptual model, which has been adopted as a factor separation approach for attributing the drivers of heat exposure (Broadbent et al. 2020), to quantify the individual and joint contributions of greenspace coverage and population growth in explaining the change of greenspace exposure inequality measured by the Gini index. As shown in **Fig. 3**, we assumed the impact of greenspace on the Gini change includes two parts: individual effect (region I, equivalent to **[a]** in **Fig. R7**) and joint effects from greenspace and population (region II, equivalent to **[b]** in **Fig. R7**). The total effects of greenspace (C_{I+II}) can be calculated by varying the greenspace coverage while keeping a fixed population.

$$C_{I+II} = \Delta Gini_{green,i} = Gini(g_{i+1}, p_i) - Gini(g_i, p_i) \quad (R22)$$

where g_{i+1} and g_i are greenspace coverage at years $i+1$ and i , respectively. p_i is population at year i . $Gini$ and $\Delta Gini_{green,i}$ is the Gini index and its change induced by greenspace coverage between year i and $i+1$.

Similarly, the total effects of population on the Gini change (C_{II+III}) can be divided into two parts: individual effect (region III, equivalent to **[c]** in **Fig. R7**) and the joint effects of both greenspace and population (region II, equivalent to **[b]** in **Fig. R7**). This impact can be calculated by varying the population distribution while keeping a fixed greenspace coverage.

$$C_{II+III} = \Delta Gini_{pop,i} = Gini(g_i, p_{i+1}) - Gini(g_i, p_i) \quad (R23)$$

where p_{i+1} is population at year $i+1$. $\Delta Gini_{pop,i}$ is the Gini change caused by population between year i and $i+1$.

Thus, the total effects of greenspace and population on the Gini change ($C_{I+II+III}$) can be quantified by varying both population distribution and greenspace coverage.

$$C_{I+II+III} = \Delta Gini_{all,i} = Gini(g_{i+1}, p_{i+1}) - Gini(g_i, p_i) \quad (R24)$$

where $\Delta Gini_{all,i}$ is the Gini change caused by both greenspace and population.

By solving Eqs. (R22-24), we can quantify individual and shared contributions from greenspace and population to the Gini change for each year i ($i = 2000, 2001, \dots, 2017, 2018$).

$$C_I = C_{I+II+III} - C_{II+III} \quad (R25)$$

$$C_{III} = C_{I+II+III} - C_{I+II} \quad (R26)$$

$$C_{II} = C_{I+II} + C_{II+III} - C_{I+II+III} \quad (R27)$$

In addition to this Venn conceptual model, we also adopted another approach proposed by Tuholske et al. (2020), to empirically quantify the comparative contributions of greenspace and population to the change of greenspace exposure inequality (i.e., the Gini change). This approach assumes that the total inequality trend β_{expo} can be linearly decomposed into the contributions of greenspace coverage β_{green} and population distribution β_{pop} (i.e., $\beta_{expo} = \beta_{green} + \beta_{pop}$), which calculation includes four major steps:

- (1) Calculate the city-level monotonic trend of greenspace exposure inequality β_{expo} with varying greenspace coverage and population distribution between 2000 and 2018.
- (2) Calculate the share of greenspace exposure inequality from greenspace coverage β_{green} by varying greenspace coverage across year and fixed population at baseline year 2000.
- (3) Calculate the share of greenspace exposure inequality trend from population growth β_{pop} by subtracting greenspace contribution from the total trend of greenspace exposure inequality, i.e., $\beta_{pop} = \beta_{expo} - \beta_{green}$. In stpes. (2-3), we can also first calculate the share of greenspace exposure inequality from population growth β_{pop} and derive the share from greenspace coverage β_{green} based on the linear model assumption, i.e., $\beta_{green} = \beta_{expo} - \beta_{pop}$.
- (4) Normalize the relative contributions as comparative contribution (CC): $CC = (|\beta_{pop}| - |\beta_{green}|) \div |\beta_{expo}|$, where $|\cdot|$ represents the absolute function.

Results show that greenspace dominates the overall change in the Gini index based on the comparative contribution (CC) metric compared to population growth (**Fig. S15**). We have included the description and attribution results of Tuholske et al. (2020) approach in the revised manuscript, which are duplicated as follows.

(Page 37, Lines 765-775): *“In addition to the Venn conceptual model, we adopted another empirical approach ⁴² to quantify the comparative contributions of greenspace and population to the change of human greenspace exposure inequality, with four major steps: 1) calculation of the monotonic trend of greenspace exposure inequality β_{expo} by varying greenspace coverage and population between 2000 and 2018; 2) calculation of greenspace coverage β_{green} (or population growth, β_{pop}) contribution with the monotonic trend analysis by varying greenspace coverage (or population) while keeping a fixed population (or greenspace coverage) at baseline year 2000; 3) calculation of population growth, β_{pop} (or greenspace coverage β_{green}) contribution by subtracting greenspace (population) contribution from the overall trend of greenspace exposure inequality, i.e., $\beta_{pop} = \beta_{expo} - \beta_{green}$ (or $\beta_{green} = \beta_{expo} - \beta_{pop}$); 4) calculation of the comparative contribution (CC) as: $CC = (|\beta_{pop}| - |\beta_{green}|) \div |\beta_{expo}|$, where $|\cdot|$ represents the absolute function.”*

(Pages 9-10, Lines 195-200): *“In addition to results based on the Gini index, our*

sensitivity analysis using the Atkinson and Theil metrics also show very similar attribution results (**Supplementary Figs. 13-14**). We also incorporated the comparative attribution approach⁴² to verify the robustness of the proposed Venn conceptual model, similar results show that the contribution from greenspace dominates the overall change in the Gini index compared to population growth (**Supplementary Fig. S15**.)”

Fig. 3. The Venn conceptual model of quantifying individual and joint contributions from greenspace and population to the overall change in the Gini index. Green circle (regions I and II) denotes the Gini index change $\Delta Gini_{green,i}$ induced by the change of greenspace provision from g_i to g_{i+1} with a fixed population p_i for year i ($i=2001, \dots, 2018$). Orange circle (regions II and III) denotes the Gini index change $\Delta Gini_{pop,i}$ induced by the population change from p_i to p_{i+1} with a fixed greenspace g_i for year i . These two circles (regions I, II, and III) denote the overall change of Gini index $\Delta Gini_{all,i}$ induced by both greenspace and population change from (g_i, p_i) to (g_{i+1}, p_{i+1}) for year i . The overlapped region II denotes the joint effects of greenspace and population changes.

Fig. S15. Comparative contributions of greenspace and population for the

temporal change in the Gini index using the empirical approach proposed by Tuholske et al. (2020). a-b. Spatial patterns of population versus greenspace for the overall change of the Gini index using the comparative contribution (CC) metric: $CC = (|\beta_{pop}| - |\beta_{green}|) \div |\beta_{expo}|$. **c-d.** City statistics from greenspace and population to the temporal change in greenspace exposure inequality. **a** and **c.** The overall trend β_{expo} and the share of greenspace β_{green} are first calculated and then the share of population is quantified: $\beta_{pop} = \beta_{expo} - \beta_{green}$. **b** and **d.** The overall trend β_{expo} and the share of population β_{pop} are first calculated and then the share of greenspace is quantified: $\beta_{green} = \beta_{expo} - \beta_{pop}$.

References:

- [1] Borcard, D., Legendre, P., & Drapeau, P. (1992). Partialling out the spatial component of ecological variation. *Ecology*, 73(3), 1045-1055.
- [2] Broadbent, A. M., Krayenhoff, E. S., & Georgescu, M. (2020). The motley drivers of heat and cold exposure in 21st century US cities. *Proceedings of the National Academy of Sciences*, 117(35), 21108-21117.
- [3] Delgado-Baquerizo, M., Reich, P. B., Bardgett, R. D., Eldridge, D. J., Lambers, H., Wardle, D. A., ... & Fierer, N. (2020). The influence of soil age on ecosystem structure and function across biomes. *Nature Communications*, 11(1), 4721.
- [4] Legendre, P., Borcard, D., & Roberts, D. W. (2012). Variation partitioning involving orthogonal spatial eigenfunction submodels. *Ecology*, 93(5), 1234-1240.
- [5] Peres-Neto, P. R., Legendre, P., Dray, S., & Borcard, D. (2006). Variation partitioning of species data matrices: estimation and comparison of fractions. *Ecology*, 87(10), 2614-2625.
- [6] Tuholske, C., Caylor, K., Funk, C., Verdin, A., Sweeney, S., Grace, K., ... & Evans, T. (2021). Global urban population exposure to extreme heat. *Proceedings of the National Academy of Sciences*, 118(41), e2024792118.

Results and Discussions:

- The structuring of this whole section makes it difficult to read. Please consider reorganizing the whole section. Include subheadings that breakup into smaller, more manageable sections. Additionally, follow a similar flow to the methods section, as this would enhance the readability of the manuscript.

> Thanks for your good question. Following your suggestion, we have separated the result section into three subsections with subheadings: (1) Spatiotemporal patterns of physical greenspace coverage; (2) Spatiotemporal patterns of greenspace exposure and associated inequality; and (3) Drivers of changing inequality in greenspace exposure.

Similarly, we also divided the method section into nine subsections with subheadings:

- (1) Research design
- (2) Global urban areas
- (3) Landsat satellite imagery
- (4) Population data
- (5) Greenspace coverage
- (6) Greenspace exposure
- (7) Greenspace exposure inequality
- (8) Temporal trend analysis
- (9) Attribution of changing inequality in greenspace exposure

Besides the subheadings, we also set the first sentence of each paragraph to express the central clause to improve the readability of the manuscript.

- These sections contain significant amount of ambiguous language, which can make it challenging to follow the authors' arguments. The authors are encouraged to clarify their language to make their points more precise and understandable.

> Thanks for your good question. We have simplified and polished the language of each sentence in the result section and the key message of each paragraph can be summarized: (1) Spatiotemporal pattern of physical greenspace coverage; (2) Spatiotemporal patterns of greenspace exposure and associated inequality; and (3) Drivers of changing inequality in greenspace exposure. We have included these subheadings to improve the readability of the result section. For each section, we also polish the language with precise and concise expressions and words.

- What is this factor of 0.001?

> Thanks. It's the magnitude level of temporal trends of greenspace coverage, human exposure, and the associated inequality. As greenspace coverage has a small range of 0-1, its magnitude level of temporal trend is relatively small.

- Line 117: Does largest temporal trend mean highest magnitude of change during the time period? Please be concise.

> Thanks. Your understanding is right. We have revised this sentence (Page 7, Lines 138-140). Now it reads:

“By continent, European cities have the largest level of change trend for greenspace exposure ($4.41 \times 0.001 \text{ yr}^{-1}$), which is twice that of Asian ($-2.11 \times 0.001 \text{ yr}^{-1}$) and Australian and Oceanian ($2.16 \times 0.001 \text{ yr}^{-1}$) cities.”

- Line 132: Unclear how greenspace supply reflects demand.

> Thanks for your good question. Physical greenspace coverage reflects the

capacity of urban greenspace to supply humans' need of greenspace environment that are used. Supply is measured as stock (detectable from satellite data). Demand is assumed by adding information about the proximity of humans, on the assumption that humans will use proximal green space (for a range of uses). Based on your and the 2nd reviewer's comment, we have replaced "demand" as "assumed demand" (Page 8, Lines 155-157). Now it reads:

*"To adjust gross greenspace supply to reflect **assumed demand**, we used population distribution maps and a population-weighted exposure framework to quantify the temporal change of human greenspace exposure and inequality."*

• Line 135: Should it not be Fig 1c or 1d?

> Thank you for pointing out this issue. It should be **Fig. 1c**. We have corrected this figure citation in the revised manuscript (Page 8, Lines 158-159).

• A clear explanation of the turning point year of 2011 in the discussion section is missing. The authors should consider providing more context and potential reasons for any significant changes or trends observed in the data around this time.

> Thanks for your good question. The turning point year of 2011 might be explained by two reasons (Zhang et al. 2021). First, the global urban population exceeded the rural population around year 2010. Second, both satellite observations and land-surface model simulations reveal that several natural and anthropogenic factors, including CO₂ fertilization, climate change, land-use change, and nitrogen deposition, caused a net global vegetation greening since 2010 (Liu et al. 2022; Piao et al. 2020), which simultaneously results into an increase in urban greenspace. We included these two potential explanations in the revised manuscript (Page 12, Lines 257-261), which is duplicated as follows.

"Interestingly, a turning point in the trend of greenspace provision and exposure was observed around the year 2011, when the world's urban population exceeded the rural population⁵⁷ and natural and anthropogenic activities plausibly related to a net global vegetation greening (e.g., CO₂ fertilization, climate change, land-use change, and nitrogen deposition) have been enhanced⁴⁸"

References:

- [1] Liu, Y., Chai, Y., Yue, Y., Huang, Y., Yang, Y., Zhu, B., ... & Ullah, W. (2022). Effects of global greening phenomenon on water sustainability. *Catena*, 208, 105732.
- [2] Piao, S., Wang, X., Park, T., Chen, C., Lian, X. U., He, Y., ... & Myneni, R. B. (2020). Characteristics, drivers and feedbacks of global greening. *Nature Reviews Earth & Environment*, 1(1), 14-27.
- [3] Zhang, X., Brandt, M., Tong, X., Ciais, P., Yue, Y., Xiao, X., ... & Fensholt, R.

(2022). A large but transient carbon sink from urbanization and rural depopulation in China. *Nature Sustainability*, 5(4), 321-328.

REVIEWERS' COMMENTS

Reviewer #1 (Remarks to the Author):

Thank you for answering all my questions in great detail. I am satisfied with the provided answers and find the paper fit for publication.

Reviewer #2 (Remarks to the Author):

The authors have carefully and comprehensively addressed my earlier comments and I am satisfied with their responses and associated revisions to the manuscript. In particular, I believe the addition of a paragraph describing the study limitations improves the quality of the paper.

Some minor grammatical and spelling errors remain in the manuscript, especially in areas where new text has been added (e.g., lines 368-371), that the authors should fix prior to publication. I leave it to the authors and editorial team to identify and correct these errors.

I recommend accepting this paper, contingent on a copy edit to correct minor errors and ensure clear communication of the findings and their implications.

Reviewer #4 (Remarks to the Author):

Dear authorship,

I greatly appreciate your careful revision of your valuable research. Very well done in general.

Some tiny aspects slipped out of sight, however:

In the letter to Reviewer 3 (pg. 32 in Letter to Reviewers) you explain so nicely the definition of greenspace exposure. This is really important to understand why you undertook this study. And you add to References Jarvis et al.; Reid et al. - however in the text this paragraph is not found. For this reason, questions remain what you understand by the terminology greenspace exposure. Beyond, the reference from the letter is not found in the manuscript. Please add.

Language:

(1) throughout the text make sure you say
either between 2000 and 2011
or from 2000 to 2011
a mixture is not possible.

(2) decide which spelling you choose
- greenspace
- green space

please be consistent throughout the text.

I would advise you to use less self-citation. This is not a good style in the highly esteemed research community.

Still, greatly appreciated to read your research and all the well-done amendments.

Response Letter

> We appreciate the editor and three reviewers' assessment of our manuscript. The detailed comments and constructive suggestions have greatly improved our study. We have well addressed all raised points and revised the manuscript carefully.

> Please find our point-by-point response below.

Reviewer #1 (Remarks to the Author):

Thank you for answering all my questions in great detail. I am satisfied with the provided answers and find the paper fit for publication.

> Thank you.

Reviewer #2 (Remarks to the Author):

The authors have carefully and comprehensively addressed my earlier comments and I am satisfied with their responses and associated revisions to the manuscript. In particular, I believe the addition of a paragraph describing the study limitations improves the quality of the paper.

Some minor grammatical and spelling errors remain in the manuscript, especially in areas where new text has been added (e.g., lines 368-371), that the authors should fix prior to publication. I leave it to the authors and editorial team to identify and correct these errors.

I recommend accepting this paper, contingent on a copy edit to correct minor errors and ensure clear communication of the findings and their implications.

> Thank you for your comments and suggestions in the first round, which greatly help improve our manuscript.

Based on your and Editor's suggestions on grammatical and spelling issues, we have carefully revised the manuscript language to better present the findings and implications from this study.

Reviewer #3 (Remarks to the Author):

Dear authorship,

I greatly appreciate your careful revision of your valuable research. Very well done in general. Some tiny aspects slipped out of sight, however:

In the letter to Reviewer 3 (pg. 32 in Letter to Reviewers) you explain so nicely the definition of greenspace exposure. This is really important to understand why you undertook this study. And you add to References Jarvis et al.; Reid et al. - however in the text this paragraph is not found. For this reason, questions remain what you understand by the terminology greenspace exposure. Beyond, the reference from the letter is not found in the manuscript. Please add.

> Thank you for your detailed assessment of our revised manuscript. Following your suggestions, we included the definition of greenspace exposure in the revised manuscript and added the related references from Jarvis et al. 2020 and Reid et al. 2018 as refs 18-19, which is depicted as follows:

“Human exposure to greenspace, measured as the averaged amount of greenspace coverage within people’s nearby environment expressed as a percentage¹⁸⁻¹⁹, has also been quantified in sampled cities²⁰⁻²².”

Language:

(1) throughout the text make sure you say either between 2000 and 2011 or from 2000 to 2011; a mixture is not possible.

> Thank you for spotting this. We have used the consistent expressions “from xx to yy” for year ranges in the revised manuscript.

(2) decide which spelling you choose

- greenspace
- green space

Please be consistent throughout the text.

> Thank you for pointing this out. We now used the consistent “greenspace” in the revised manuscript.

I would advise you to use less self-citation. This is not a good style in the highly esteemed research community.

> Thank you. Actually, the included literature from our previous studies are all highly relevant to the present one. To address this issue, we have excluded the three following references from our manuscript.

- 15 Song Y, Huang B, Cai J, Chen B. Dynamic assessments of population exposure to urban greenspace using multi-source big data. *Science of the Total Environment* **634**, 1315-1325 (2018).
- 16 Song Y, Chen B, Kwan M-P. How does urban expansion impact people's exposure to green environments? A comparative study of 290 Chinese cities. *Journal of Cleaner Production* **246**, 119018 (2020).
- 17 Tu Y, Chen B, Yang J, Xu B. Olympic effects on reshaping urban greenspace of host cities. *Landscape and Urban Planning* **230**, 104615 (2023).

Still, greatly appreciated to read your research and all the well-done amendments.

> Thank you again for your positive comment on our revised manuscript.